# Causal network models of SARS-CoV-2 expression and aging to identify candidates for drug repurposing

Anastasiya Belyaeva[1,5], Louis Cammarata[2,5], Adityanarayanan Radhakrishnan[1,5], Chandler Squires[1], Karren Dai Yang[1], G. V. Shivashankar[3,4] & Caroline Uhler [1✉]

Given the severity of the SARS-CoV-2 pandemic, a major challenge is to rapidly repurpose existing approved drugs for clinical interventions. While a number of data-driven and experimental approaches have been suggested in the context of drug repurposing, a platform that systematically integrates available transcriptomic, proteomic and structural data is missing. More importantly, given that SARS-CoV-2 pathogenicity is highly age-dependent, it is critical to integrate aging signatures into drug discovery platforms. We here take advantage of large-scale transcriptional drug screens combined with RNA-seq data of the lung epithelium with SARS-CoV-2 infection as well as the aging lung. To identify robust druggable protein targets, we propose a principled causal framework that makes use of multiple data modalities. Our analysis highlights the importance of serine/threonine and tyrosine kinases as potential targets that intersect the SARS-CoV-2 and aging pathways. By integrating transcriptomic, proteomic and structural data that is available for many diseases, our drug discovery platform is broadly applicable. Rigorous in vitro experiments as well as clinical trials are needed to validate the identified candidate drugs.

[1] Massachusetts Institute of Technology, Cambridge, MA, USA. [2] Harvard University, Cambridge, MA, USA. [3] ETH Zurich, Zurich, Switzerland. [4] Paul Scherrer Institute, Villigen, Switzerland. [5] These authors contributed equally: Anastasiya Belyaeva, Louis Cammarata, Adityanarayanan Radhakrishnan. ✉email: cuhler@mit.edu

Candidates for drug repurposing have mainly been identified based on an understanding of their pharmacology or based on retrospective analyses of their clinical effects. Recently, also more systematic computational methods combined with large-scale experimental screens have been employed[1]. The Connectivity Map (CMap) containing gene-expression profiles generated by dosing thousands of small molecules, including many Food and Drug Administration (FDA) approved compounds, in a number of human cell lines has been particularly valuable in this regard[2]. Common computational approaches include signature matching, where the signature of a drug is determined for example using CMap and compared to the reverse signature of a disease to identify drugs with high correlation[3]. In addition, approaches to identify drug or disease networks based on known pathways, protein–protein interactions, gene expression, or genome-wide association studies have also been employed[4–6]. To capitalize on the abundance of data, it is critical to develop computational platforms that can integrate different data modalities, including gene expression, drug targets, and signatures, as well as protein–protein interactions. In addition, a drug represents an intervention in the system and only a causal framework allows predicting the effect of an intervention. It is, therefore, critical to capitalize on recent advances in causal inference[7,8] in particular with respect to the use of interventional data[9–12].

Given the current coronavirus disease 2019 (COVID-19) crisis, there is an urgent need for the development of robust drug repurposing methods. Coronaviruses belong to the family of positive-strand RNA-viruses. While most coronaviruses infect the upper respiratory tract and cause mild illness, they can have serious effects as exemplified by the severe acute respiratory syndrome coronavirus (SARS-CoV) epidemic and now the SARS-CoV-2 pandemic[13]. Recent studies have shown that coronaviruses use canonical inflammatory pathways (e.g., NF-$\kappa$B) of the host cell for their replication, while simultaneously dampening their outward inflammatory signaling[14,15]. This delicate partial up and downregulation of inflammatory pathways by coronaviruses has represented major challenges for therapeutic interventions[16]. While the infection rates for these viruses are similar among different age groups, the morbidity and fatality rates are significantly higher in the aging population[17,18]. The respiratory system of aging individuals is characterized by alterations of tissue stiffness[19]. Notably, recent micropatterning experiments have shown that cells subjected to substrates of different stiffness stimulated with the same cytokine (TNF-$\alpha$) exhibit different downstream NF-$\kappa$B signaling[20]. In a recent commentary, we outlined that the cross-talk between coronavirus infection and cellular aging could play a critical role in the replication of the virus in host cells by differentially intersecting with NF-$\kappa$B signaling[21]. This suggests that efforts for drug repurposing should analyze SARS-CoV-2 infected host cell expression programs in conjunction with aging-dependent programs. While a number of studies are underway that investigate viral integration/replication and interactions with the host cell[6,22], to our knowledge the interplay of SARS-CoV-2 host response and aging has not been explored in the context of drug development and repurposing.

In this paper, we propose a computational platform for drug repurposing, which integrates transcriptomic, proteomic, and structural data with a principled causal framework, and we apply it in the context of SARS-CoV-2 (Fig. 1, Supplementary Fig. 1). Given the age-dependent pathogenicity of SARS-CoV-2, we first identify genes that are differentially regulated by SARS-CoV-2 infection and aging based on bulk RNA-seq data from[23,24]. We then use an autoencoder, a type of artificial neural network used to learn data representations in an unsupervised manner[25,26],

to embed the CMap data together with the SARS-CoV-2 expression data for signature matching to obtain an ordered list of FDA-approved drugs. In particular, we show that over-parameterized autoencoders align drug signatures from different cell types and thus allow constructing synthetic interventions[27,28] by translating the effect of a drug from one cell type to another. We then construct a combined SARS-CoV-2 and aging interactome using a Steiner tree analysis to connect the differentially expressed genes within a protein–protein interaction network[29,30]. By intersecting the resulting combined SARS-CoV-2 and aging interactome with the targets of the top-ranked FDA-approved drugs from the previous analysis, we identify serine/threonine and tyrosine kinases as potential drug targets for therapeutic interventions. Causal structure discovery methods applied to the combined SARS-CoV-2 and aging interactome show that the identified protein kinase inhibitors such as axitinib, dasatinib, pazopanib, and sunitinib target proteins that are upstream from genes that are differentially expressed in SARS-CoV-2 infection and aging, thereby validating these drugs as being of particular interest for the repurposing against COVID-19, postinfection. While we apply our computational platform in the context of SARS-CoV-2, our algorithms integrate data modalities that are available for many diseases, thereby making them broadly applicable.

## Results
**Differential expression analysis identifies genes that intersect the SARS-CoV-2 host response and aging pathways.** Since age is strongly associated with severe outcomes in patients with COVID-19, we sought to analyze genes differentially expressed in normal versus SARS-CoV-2-infected cells as well as genes differentially expressed in young versus old individuals. Used as a model system for lung epithelial cells and the effect of SARS-CoV-2 infection, we obtained from ref. [23], RNA-seq samples from normal and SARS-CoV-2-infected A549 lung alveolar cells as well as A549 cells supplemented with ACE2 (A549-ACE2), a receptor that has been shown to be critical for SARS-CoV-2 cell entry[31]. Fig. 2a shows the expression of A549-ACE2 cells infected with SARS-CoV-2 in comparison to normal A549-ACE2 cells, with many genes upregulated as a result of the infection, as expected. Given the availability of A549 data with/without ACE2 and with/without SARS-CoV-2 infection, we removed genes from this initial list of differentially expressed genes that were just ACE2-specific or just SARS-CoV-2 infection-specific to extract a more refined expression pattern of ACE2-mediated SARS-CoV-2 infection ("Methods", Fig. 2b). The rationale was to remove genes linked to the response of the ACE2 receptor to signals other than SARS-CoV-2 infection or genes involved in the entry of SARS-CoV-2 into the cell through means other than the ACE2 receptor, which has been shown to be the critical mode of entry in humans[31]. Gene ontology (GO) enrichment analysis revealed enrichment in a mitotic cell cycle as the top term, further supporting the removal of these genes (Supplementary Fig. 2). The remaining 1926 genes are denoted in red in Fig. 2a, b and are used for the subsequent analysis. GO enrichment analysis of these genes revealed that they are significantly enriched in the type I interferon signaling pathway and defense response to the virus in addition to other GO terms (Fig. 2c). Next, in order to analyze the link between SARS-CoV-2 infection and aging, we analyzed RNA-seq samples from the lung of different aged individuals collected as part of the Genotype-Tissue Expression (GTEx) study[24]. Given the stark increase in case fatality rates of COVID-19 after age 70[17,18], we performed a differential expression analysis comparing the youngest group (20–29 years old) and oldest group (70–79 years old), thereby identifying 1923

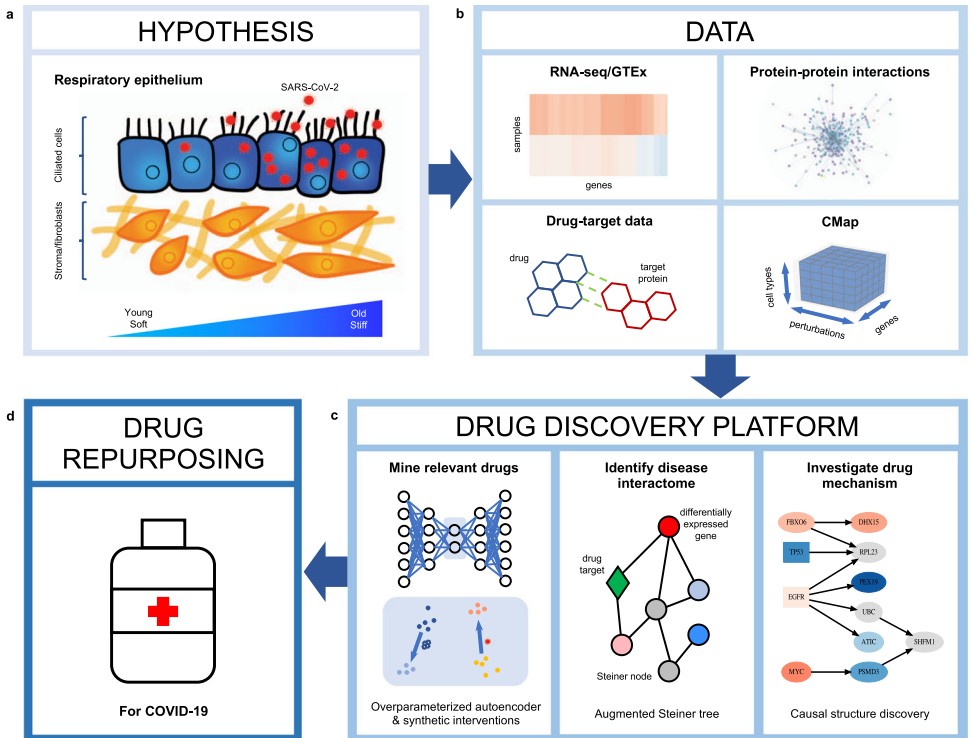

**Fig. 1 Overview of computational drug repurposing platform for COVID-19. a** COVID-19 is associated with more severe outcomes in older individuals, suggesting that gene expression programs associated with SARS-CoV-2 and aging must be analyzed in tandem. A potential hypothesis regarding the cross-talk between SARS-CoV-2 and aging relies on changes in tissue stiffness in older individuals, outlined in ref. [21]. Ciliated cells are denoted in blue, stromal/fibroblast cells are in orange and SARS-CoV-2 viral particles are in red. **b** In order to identify potential drug candidates for COVID-19, we integrated RNA-seq data from SARS-CoV-2-infected cells, obtained from ref. [23], and RNA-seq data from the lung tissue of young and old individuals, collected as part of the Genotype-Tissue Expression (GTEx) project[24], with protein–protein interaction data (from ref. [42]), drug–target data (from DrugCentral[45]) and the large-scale transcriptional drug screen Connectivity Map (CMap)[2]. **c, d** Based on this data, we develop a drug repurposing pipeline, which consists of first, mining relevant drugs by matching their signatures with the reverse disease signature in the latent embedding obtained by an overparameterized autoencoder and sharing data across cell types to obtain missing drug signatures via synthetic interventions. Blue and orange points in the latent space represent data associated with the drug screen and the SARS-CoV-2 infection study. Second, we identify a disease interactome within the protein–protein interaction network by identifying a minimal subnetwork that connects the genes differentially expressed by SARS-CoV-2 infection and aging using a Steiner tree analysis. Third, we validate the drugs identified in the first step that have targets in the interactome (greed diamond) by identifying the potential drug mechanism using causal structure discovery. Nodes are colored according to the $\log_2$-fold gene expression change associated with SARS-CoV-2 infection, and gray nodes indicate Steiner nodes.

genes differentially regulated in aging (Fig. 2d, Supplementary Fig. 3). As shown in Fig. 2e, these genes show a significant overlap with the 1926 genes found to be differentially regulated by SARS-CoV-2 ($p$ value = 0.01999, one-sided Fisher's exact test), thereby confirming results obtained using a different analysis in ref. [32]. Interestingly, these 219 genes that we found to intersect the SARS-CoV-2 infection and aging pathways (Fig. 2e) display concordant changes in gene expression (i.e., the majority of genes are either upregulated or downregulated with SARS-CoV-2 infection and aging) as shown by the $\log_2$-fold changes in Fig. 2f and Supplementary Fig. 4a. The association in the directionality of regulation between SARS-CoV-2 infection and aging is statistically significant ($p$ value $< 2.2 \times 10^{-16}$, one-sided Fisher's exact test), thereby providing further evidence for the interplay of SARS-CoV-2 host response and aging as hypothesized in ref. [21]. Fig. 2g shows the $\log_2$-fold changes of the ten most differentially expressed genes across aging and SARS-CoV-2 infection (based on the sum of their ranks with Supplementary Fig. 4b showing the distribution of the ranks).

**Identification of SARS-CoV-2 infection signature in reduced L1000 gene expression space.** Next, we focused our analysis on

identifying the SARS-CoV-2 transcriptional signature, which we then correlated with the transcriptional signatures of FDA-approved drugs in CMap to identify drugs that could revert the effect of SARS-CoV-2 infection. While this analysis resulting in an initial ranking of FDA-approved drugs did not take the transcriptional signature of aging into account, aging was a critical component in the final selection of FDA-approved drugs described below.

Since gene expression in CMap was quantified using L1000 reduced representation expression profiling[2], which measures gene expression of 1000 representative genes, we first sought to analyze whether these genes sufficiently capture the transcriptional signature of SARS-CoV-2 infection. For this, we intersected the genes measured both by Blanco et al.[23] and CMap[2], resulting in 911 genes. We found a statistically significant overlap between the genes identified as differentially expressed by SARS-CoV-2 infection in Fig. 2 and the L1000 genes ($p$ value = $7.94 \times 10^{-16}$, one-sided Fisher's exact test), thereby providing a rationale for using the CMap database for drug identification in this disease context (Fig. 3a). We thus proceeded to obtain the signature of SARS-CoV-2 infection in the reduced L1000 gene expression space by projecting the RNA-seq data of A549 cells with and without ACE2 receptor and SARS-CoV-2 infection onto the

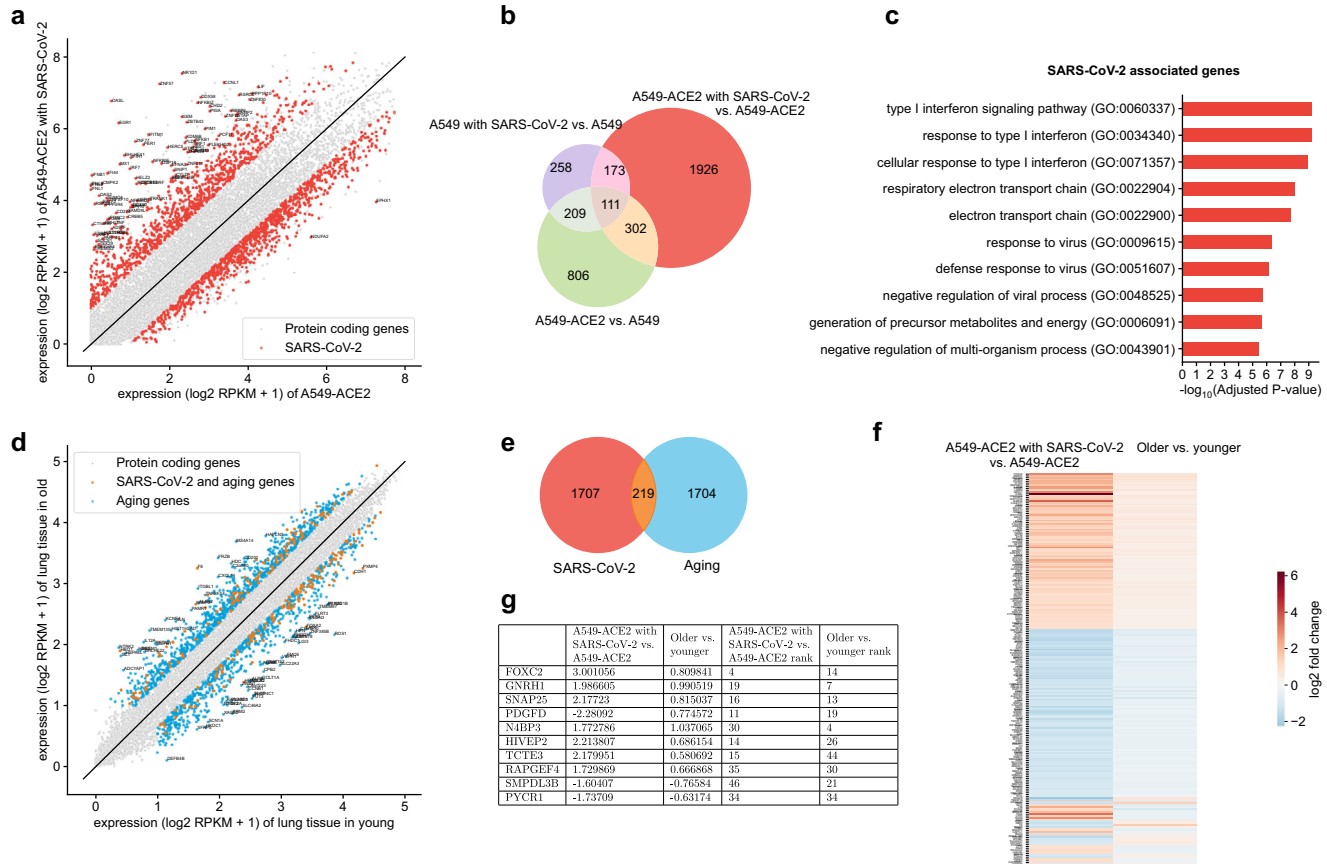

**Fig. 2 Identification of differentially regulated genes in SARS-CoV-2 infection and aging. a** Gene expression ($\log_2$ RPKM + 1) of A549-ACE2 cells infected with SARS-CoV-2 versus normal A549-ACE2 cells. Genes associated with ACE2-mediated SARS-CoV-2 infection after removing just ACE2-specific or just SARS-CoV-2 infection-specific genes are shown in red with the remaining protein-coding genes shown in gray. **b** Venn diagram, showing the number of genes in sets considered for obtaining the 1926 genes in the red subset and shown in red in (**a**) associated with ACE2-mediated SARS-CoV-2 infection. Purple, red, and green whole circles indicate differentially expressed genes associated with A549 cells infected with SARS-CoV-2, A549-ACE2 cells infected with SARS-CoV-2, and A549 cells with and without ACE2, respectively. **c** Top ten gene ontology terms associated with SARS-CoV-2 infection (adjusted $p$ value < 0.05, Benjamini–Hochberg procedure). **d** Gene expression ($\log_2$ RPKM + 1) of cells collected from lung tissue of older (70–79 years old) versus younger (20–29 years old) individuals. Differentially expressed genes associated with aging are shown in blue and genes that are associated with both aging and SARS-CoV-2 are shown in orange with the remaining protein-coding genes shown in gray. **e** Venn diagram of genes associated with SARS-CoV-2 (red circle) and aging (blue circle); intersection (orange) is significant ($p$ value = 0.01999, one-sided Fisher's exact test). **f** Heatmap of $\log_2$-fold changes of differentially expressed genes shared by SARS-CoV-2 and aging; most genes show concordant expression, i.e., they are both upregulated or both downregulated with SARS-CoV-2 infection and aging. **g** Table of the top ten most differentially expressed genes across aging and SARS-CoV-2, based on the sum of their ranks with $\log_2$-fold changes for each gene.

shared 911 genes. The resulting signatures of SARS-CoV-2 infection and ACE2 receptor are visualized using the first two principal components in Fig. 3b. Interestingly, the signature of SARS-CoV-2 infection (indicated by arrows) was aligned across both A549 and A549-ACE2 cells as well as across different levels of infection (MOI of 0.2 and 2), suggesting that the SARS-CoV-2 transcriptional signature was captured robustly by the L1000 genes, thus providing further rationale for using CMap to identify drugs that could reverse the effect of SARS-CoV-2 infection.

**Combined autoencoder and synthetic interventions framework to identify drug signatures and rank FDA-approved drugs for SARS-CoV-2.** Next, we sought to determine transcriptional drug signatures using the CMap database, which includes among other cell lines A549. The data were visualized using Uniform Manifold Approximation and Projection (UMAP)[33] in Supplementary Fig. 5a, showing that the perturbations clustered by cell type and hence the drug signatures were small relative to the differences between cell types. We intersected the perturbations from CMap

with a list of FDA-approved drugs using Slinky[34], resulting in 759 drugs of which 605 were available for A549. After removing batch effects using k-means clustering (see "Methods" and Supplementary Fig. 5b), we computed initial signatures of these drugs based on the mean before and after drug perturbation in A549 cells. Fig. 3c shows a selection of drug signatures in relation to the signature of SARS-CoV-2 infection visualized using the top two principal components.

Since the effect of a drug can be cell type-specific[35], this standard approach to computing drug signatures may not allow extrapolating the obtained signatures beyond A549 cells. In order to determine robust drug signatures and consider also FDA-approved drugs that have been dosed on cell lines other than A549 in CMap, we employed an autoencoder framework. Autoencoders, a particular class of neural networks where input is mapped through a latent space to itself, have been widely used for representation learning[25,26,36] and more recently also in genomics and single-cell biology[37–39]. We trained an autoencoder (architecture described in Supplementary Fig. 6) to minimize reconstruction error on CMap data and applied it to data from

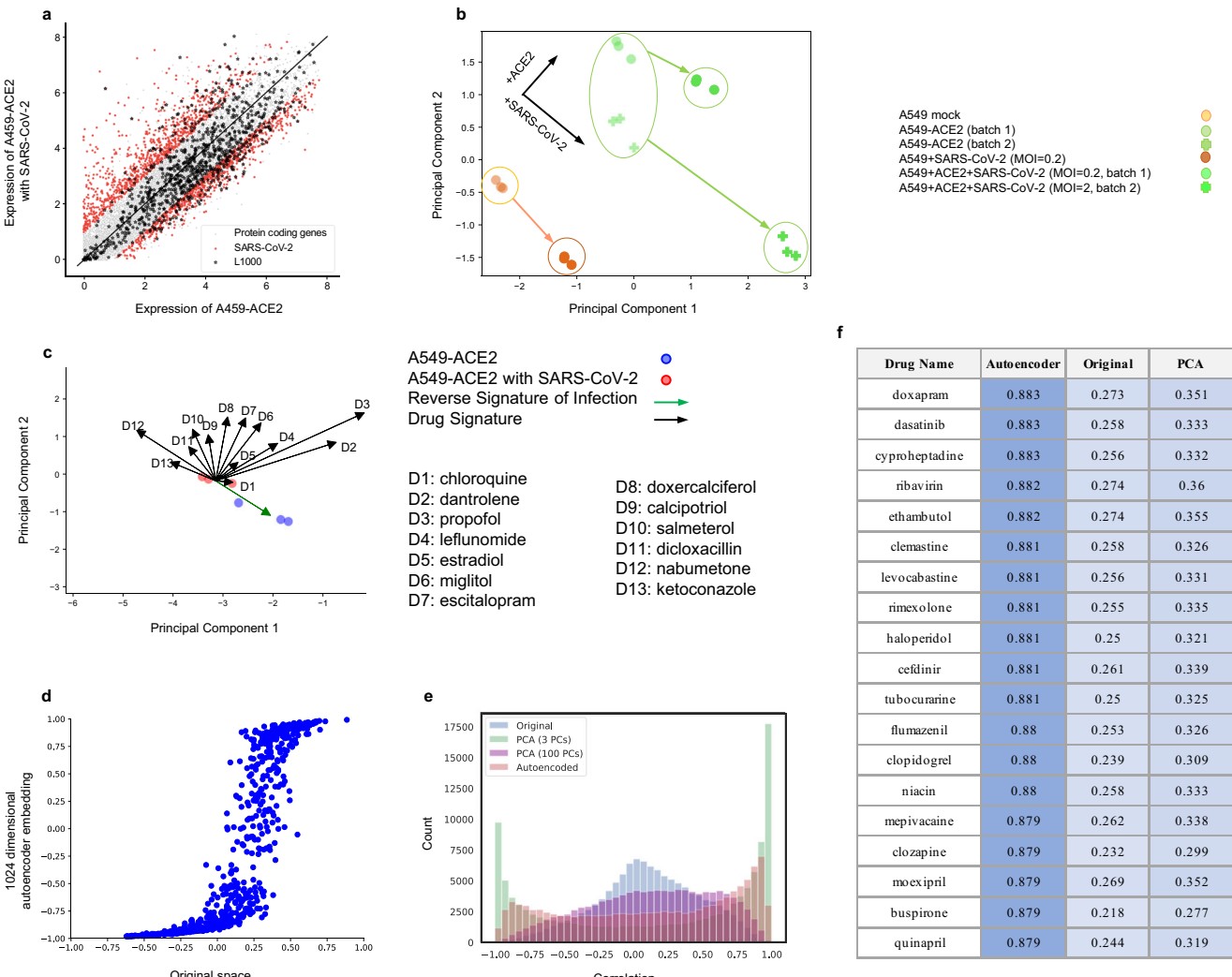

**Fig. 3 Mining FDA-approved drugs by correlating disease and drug signatures using an overparameterized autoencoder embedding. a** Gene expression ($\log_2$ RPKM + 1) of A549-ACE2 cells infected with SARS-CoV-2 versus normal A549-ACE2 cells with genes collected as part of the CMap study using the L1000 reduced representation expression profiling method highlighted as stars, showing that L1000 genes significantly overlap with SARS-CoV-2 associated genes, shown in red, (p value = $7.94 \times 10^{-16}$, one-sided Fisher's exact test). **b** Signature of SARS-CoV-2 infection on A549 and A549-ACE2 cells visualized using the first two principal components based on RNA-seq data from ref. [23]. The signature of SARS-CoV-2 infection is aligned across normal A549 and A549-ACE2 cells as well as across different levels of infection. Green and orange points indicate data from A549-ACE2 and A549 cells, respectively. Circles and crosses indicate data from two different batches, the multiplicity of infection (MOI) of 0.2 versus 2, respectively. **c** Comparison of the signatures of a selection of 13 representative FDA-approved drugs (black arrows) as compared to the reverse signature of SARS-CoV-2 infection based on A549-ACE2 cells (green arrow) visualized using the first two principal components. Drugs whose signatures maximally align with the direction from SARS-CoV-2-infected cells (red) to normal cells (blue) are considered candidates for treatment. As expected, drugs have varying signatures of varying magnitudes. **d** Correlation between drug signatures in A549 and MCF7 cells when using the original L1000 expression space versus the embedding obtained from an overparameterized autoencoder. The overparameterized autoencoder aligns the drug signatures in A549 and MCF7 cells by shifting the correlations towards −1 or 1 while maintaining the sign of the correlation in the original space. **e** Histogram of correlations between cell types for a given drug using original L1000 gene expression vectors (blue), overparameterized autoencoder embedding (pink), top 100 principal components (purple), and top 3 principal components (green). The overparameterized autoencoder achieves about the same alignment of drug signatures as using the top three principal components, while at the same time faithfully reconstructing the data ($10^{-7}$ training error). **f** A list of drugs whose signatures maximally align with the direction from SARS-CoV-2 infection to normal in A549-ACE2 cells (MOI 2) with respect to correlations using the overparameterized autoencoder embedding, the original L1000 gene expression space, and the top 100 principal components.

Blanco-Melo et al.[23] in the L1000 gene expression space. We then computed the disease and drug signatures based on the embedding of the data in the latent space. Interestingly, by comparing the correlations between drug signatures obtained from A549 cells and MCF7 cells (Fig. 3d) as well as HCC515 cells (Supplementary Fig. 8), cell lines with many perturbations in CMap, it is apparent that the autoencoder aligned the drug signatures across different cell types. While autoencoders and

other generative models have been used for computing signatures of perturbations also in other works[39,40], these works have used autoencoders in the standard way to obtain a lower-dimensional embedding of the data. Motivated by our recent work which, quite counter-intuitively, described various benefits of using autoencoders to learn a latent representation of the data that is higher-dimensional than the original space[41], we found that overparameterized autoencoders not only led to the better

reconstruction of the data than standardly used autoencoders (Supplementary Fig. 7 and architectures described in Supplementary Fig. 6), but also to a better alignment of drug signatures between different cell types (Supplementary Fig. 8). Interestingly, overparameterized autoencoders provided about the same alignment of drug signatures as using the top three principal components (Fig. 3e), while at the same time allowing a near-perfect reconstruction of the original gene expression vectors from the embedding. We thus used this latent space embedding to rank the drugs based on their correlation with the reverse disease signature in A549 cells (Supplementary Data 1). Since overparameterized autoencoders aligned drug signatures across cell types, this embedding also allowed constructing synthetic interventions[27,28], i.e., to predict the effect of a drug on A549 cells without measuring it, by linearly transferring the corresponding drug signature in the latent space from a cell type where it has been measured. In this way, we obtained an enlarged list of drug signatures, which we correlated in the latent space with the reverse disease signature to obtain further candidates of FDA-approved drugs for SARS-CoV-2 (Supplementary Data 1). To compare the correlations obtained with the different embeddings, a list of the top-ranked drugs is shown in Fig. 3f and the similarity between drug lists is quantitatively assessed by an analysis akin to a receiver operating characteristic (ROC) plot (Supplementary Note and Supplementary Fig. 9), showing that the drug lists obtained using an embedding in the PCA or the original space are similar but not identical to the autoencoder embedding (area under the ROC curve (AUC) of 0.901 and 0.904, respectively). Interestingly, these drug lists contain various drugs that were identified also in[6] using a different analysis (clemastine, haloperidol, ribavirin) or are currently in clinical trials (ribavirin, quinapril). To put these AUC values into perspective and assess the robustness of the identified drug list using the autoencoder embedding, we repeated the analysis on two other SARS-CoV-2 datasets from[23], namely infected A549 cells without ACE2 supplement as well as samples collected at a lower MOI (0.2 instead of 2). This resulted in very similar drug lists (Supplementary Fig. 10); in fact, the drug lists from A549 cells with and without ACE2 supplement in the autoencoder embedding were more similar than the drug lists obtained from the PCA and the original space embedding.

## Steiner tree analysis identifies candidate drug targets by constructing combined SARS-CoV-2 and aging interactome. Our differential expression analysis revealed relevant genes to investigate in the context of SARS-CoV-2 infection and aging, while the combined autoencoder and synthetic interventions analysis provided candidate FDA-approved drugs for reverting the effect of SARS-CoV-2 infection. Next, we integrated these two separate analyses to obtain a final list of FDA-approved drugs by constructing a combined SARS-CoV-2 infection and aging protein-protein interactome and intersecting it with the targets of the candidate drugs (Fig. 4a). For this, we selected the differentially expressed genes identified in Fig. 2f that showed concordant regulation between aging and SARS-CoV-2 infection and intersected them with the nodes of the human protein–protein interaction (PPI) network (IRefIndex Version 14[42]), which contains 182,002 interactions between 15,759 human proteins along with a confidence measure for each interaction. This resulted in 162 protein-coding genes, which we call terminals (Supplementary Fig. 11 and "Methods"). To gain a better understanding of the molecular pathways connecting these terminal genes, we used a Steiner tree algorithm[30,43] to determine a "minimal" subnetwork or interactome within the PPI network that connects these genes (see "Methods"). A Steiner tree is minimal in that it is a

minimum weight subnetwork that connects the terminals. As edge weights in the PPI network, we used 1 minus the confidence in the corresponding interactions so as to favor high-confidence edges. After a careful sensitivity analysis to select the various tuning parameters ("Methods" and Supplementary Fig. 12), this resulted in an interactome containing 252 nodes and 1003 edges (Fig. 4b and Supplementary Fig. 13). Interestingly, the interactome contained five genes whose corresponding proteins have been found in ref. [6] to interact with SARS-CoV-2 proteins (EXOSC5, FOXRED2, LOX, RBX1, and RIPK1). The two-nearest-neighborhoods of these proteins are shown in Fig. 4c. Another Steiner tree analysis revealed that two additional SARS-CoV-2 interaction partners (CUL2 and HDAC2) were connected to the identified interactome via few high-confidence edges (Supplementary Figs. 14–16).

Next, we intersected the interactome with the targets of the candidate drugs identified in the previous analysis. A compound was considered if its signature matched the reverse SARS-CoV-2 signature with at least a correlation of 0.86, resulting in 142 FDA-approved drugs (see "Methods"). The targets of these drugs were determined using DrugCentral[44,45] and filtered for high affinity (activity constants lower than 10 μM, a common threshold used in the field for $K_i$, $K_d$, IC50, or EC50). Interestingly, the resulting drugs, shown in Fig. 4d, consisted (with few exceptions) of protein kinase inhibitors (e.g., axitinib, dasatinib, pazopanib, and sunitinib). To analyze the specificity of our findings to SARS-CoV-2 infection in aged individuals, we repeated the above analysis without using the GTEx data. This resulted in an interactome containing 1052 edges across 270 nodes, 42 of which (15%) were also present in the interactome taking age into consideration (Supplementary Fig. 17). This pure SARS-CoV-2 interactome contained six SARS-CoV-2 interaction partners (ETFA, GNB1, NUP62, RBX1, RIPK1, and SNIP1). Drugs targeting proteins in this interactome belonged to several families including serotonin inhibitors (clozapine, cyproheptadine, desipramine, and methysergide), histamine H1 blockers (clemastine, cyproheptadine, and ketotifen), protein kinase inhibitors (including axitinib, dasatinib, pazopanib, and sunitinib) and HDAC inhibitors (vorinostat and belinostat). This analysis shows that taking aging into account acted as a valuable filter for the identification of drugs.

## Causal structure discovery methods validate serine/threonine and tyrosine kinases as critical targets in SARS-CoV-2 infection in the elderly. Finally, in order to suggest putative causal drug mechanisms and validate the predicted drugs for COVID-19, we supplemented the PPI analysis with causal structure discovery. Since the edges in the PPI network and hence in the SARS-CoV-2 and aging interactome are undirected, it is a priori not clear whether a drug that targets a node in the interactome has any effect on the differentially expressed terminal nodes, since the target may be downstream of these nodes (Fig. 5a). To understand which genes can be modulated by a drug, it is therefore critical to obtain a causal (directed) network. We obtained single-cell RNA-seq data for A549 cells from[46] and intersected it with the genes present in the combined SARS-CoV-2 and aging interactome. To learn the (causal) regulatory network among these genes, we took advantage of recently developed causal structure discovery algorithms, in particular, the greedy sparsest permutation (GSP) algorithm: it performs a greedy search over orderings of the genes to find the sparsest causal network that best fits the data, and it has been successfully applied to single-cell gene expression data before[11,12,47]. To validate the obtained causal model and benchmark the performance of GSP to other prominent causal structure discovery algorithms including

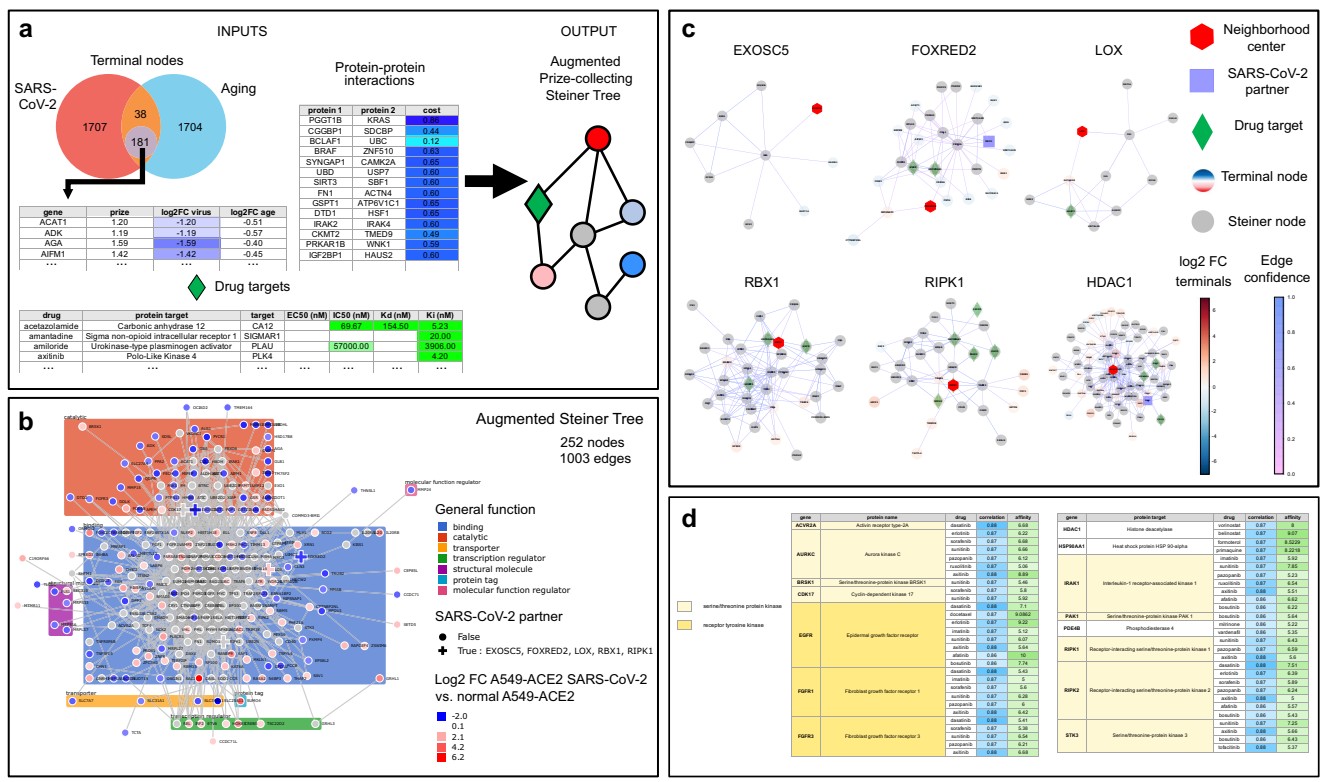

**Fig. 4 Drug target discovery via Steiner tree analysis to identify putative molecular pathways linking differentially expressed genes in SARS-CoV-2 infection and aging. a** The general procedure takes as input a list of genes of interest (terminal nodes) with prizes indicating their respective importance, a protein–protein interaction (PPI) network with edge cost/confidence information (e.g., from IRefIndex v14[42], edge cost shown by blue shading), and a list of drugs of interest along with their protein targets and available activity constants (e.g., from DrugCentral[44,45], activity constants shown by green shading). In this study, we consider 181 terminal nodes, shown as a purple circle in the Venn diagram (of which 162 are present in the PPI network) corresponding to genes differentially expressed in SARS-CoV-2 infection (red circle) and aging (blue circle) from Fig. 2 that are either upregulated in both SARS-CoV-2 infection and aging or downregulated in both SARS-CoV-2 infection and aging. The prize of a terminal node equals the absolute value of its $\log_2$-fold change in SARS-CoV-2-infected A549-ACE2 cells versus normal A549-ACE2 cells (shown in purple shading) based on the data from ref. [23]. Terminals and PPI data are processed using OmicsIntegrator2[30] to output the disease interactome, i.e., the subnetwork induced by a Steiner tree, with drug targets indicated by green diamonds and terminal nodes colored according to their prizes. Gray nodes represent Steiner nodes. **b** Interactome obtained using this procedure. Proteins are grouped by general function (colored boxes in the background) and marked with a cross if they are known to interact with SARS-CoV-2 proteins based on data from[6]. **c** 2-Nearest-neighborhoods of nodes of interest (denoted by a red hexagon) in the interactome. Proteins known to interact with SARS-CoV-2 are denoted by blue squares, drug targets are denoted as green diamonds, terminal nodes are colored according to their $\log_2$-fold change in SARS-CoV-2-infected A549-ACE2 cells versus normal A549-ACE2 cells, Steiner nodes appear in gray. Edges are colored according to edge confidence, which is thresholded to improve readability (see "Methods"). **d** Table of drug targets and corresponding drugs in the interactome. Selected drugs are FDA-approved, high affinity (at least one of the activity constants $K_i$, $K_d$, IC50 or EC50 is below 10 μM), and match the SARS-CoV-2 signature well (correlation > 0.86). The affinity column displays (and is colored by) $-\log_{10}($ activity $)$. The correlation column displays (and is colored by) correlations between drug signatures and the reverse signature of SARS-CoV-2 infection based on the overparameterized autoencoder embedding. Discovered drug targets generally fall into two categories: serine/threonine protein kinases (light yellow) and receptor tyrosine kinases (dark yellow). The remaining drug targets are in white. The protein name corresponding to each gene is included.

PC and GES[48], we took advantage of the gene knockout and overexpression data available from CMap. A causal model should allow predicting the effect of such interventions. Thus, for each such gene knockout and overexpression experiment in CMap that targeted a gene in the interactome, we inferred the genes whose expression changed as a result of the intervention, when compared to control samples ("Methods" and Supplementary Fig. 18a). We then constructed ROC curves to evaluate GSP, PC, and GES by varying their tuning parameters and counting an edge $i \rightarrow j$ as a true positive if intervening on gene $i$ resulted in a change in the expression of gene $j$ and a false positive otherwise, thereby showing that GSP exceeded random guessing based on the PPI network ($p$ value = 0.0177, see "Methods") and outperformed the other methods (Supplementary Fig. 18b).

Having established that the causal network obtained by GSP can be used to predict the effect of an intervention, we turned to analyzing the regulatory effects of the identified candidate drugs on the SARS-CoV-2 and aging interactome in A549 cells. The main connected component of the corresponding causal graph is shown in Fig. 5b (see also Supplementary Fig. 19a) highlighting the drug targets and the genes that were found to be differentially expressed by SARS-CoV-2 infection and aging. We then traced the possible downstream effects for each identified drug, thereby finding that the protein kinase inhibitors and HDAC inhibitors could target the majority of differentially expressed genes in this connected component (Supplementary Table 1). Similarly, we traced the downstream effects for each gene in the interactome that can be targeted by one of the identified drugs, thereby finding that EGFR, FGFR3, HDAC1, HSP90AA1, IRAK1, PAK1, RIPK1, RIPK2, and STK3 all have downstream nodes in the interactome with RIPK1 having the largest number of them (127). To validate these results in a broader context, we obtained single-cell

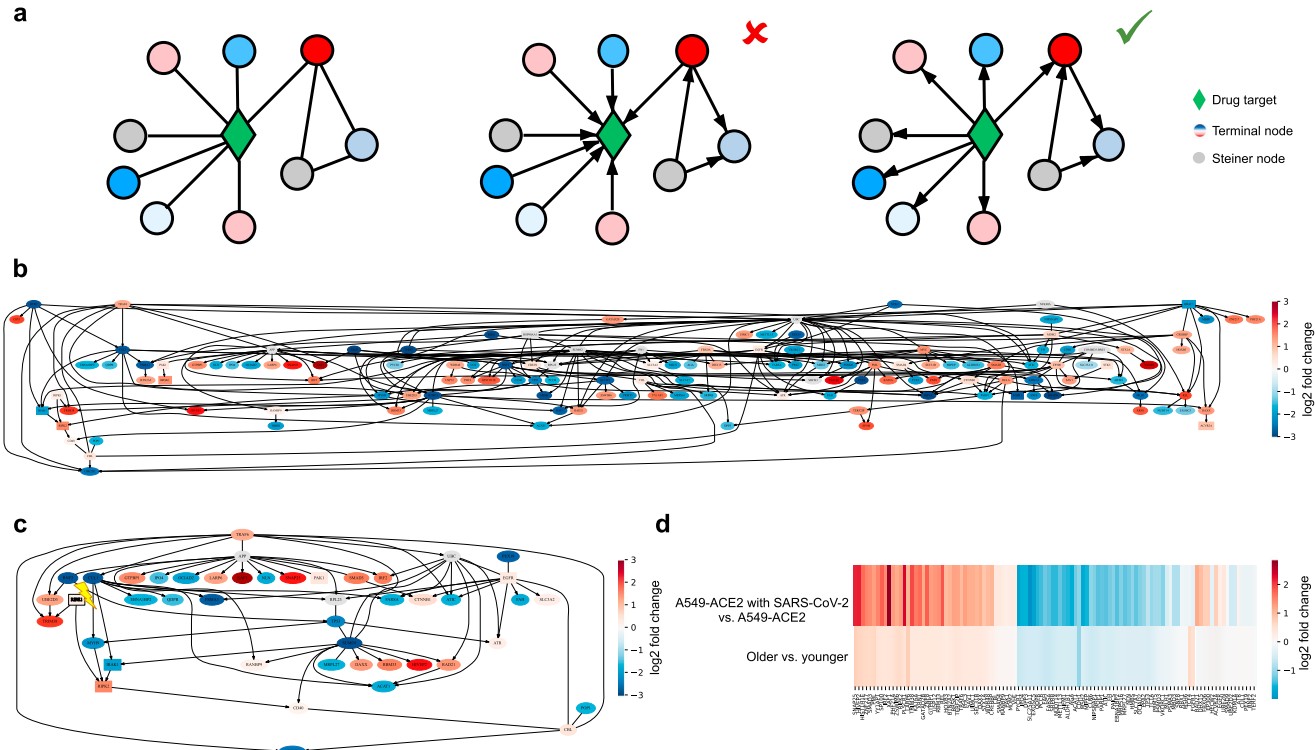

**Fig. 5 Causal mechanism discovery of potential drug targets. a** In an undirected protein–protein interaction network (left), edge directions for a particular drug target (green diamond) are unknown. Establishing causal directions is important since it is of interest to avoid drug targets that do not have many downstream nodes in the disease interactome (middle) and instead choose drug targets that have a causal effect on many downstream nodes in the disease interactome (right). **b** Causal network underlying the combined SARS-CoV-2 and aging interactome in A549 cells with gene targets of selected drugs in boxes (largest connected component shown). **c** Causal subnetwork of A549 cells corresponding to nodes within five nearest neighbors of RIPK1 (highlighted with lightning bolt). Drug targets are represented by boxes. In **a–c**, the node color corresponds to the log$_2$-fold change of expression in A549-ACE2 cells with SARS-CoV-2 infection versus without SARS-CoV-2 infection. Gray nodes represent Steiner nodes. **d** Heatmap of log$_2$-fold change of genes that are downstream of RIPK1.

RNA-seq data from ref. [49] and repeated the analysis in AT2 cells, which have been shown to be critically affected by SARS-CoV-2 in humans[31]. The resulting causal network for AT2 cells (Supplementary Fig. 19b) is similar to the one for A549 cells, intersecting it in 55.3% of the edges, with EGFR, HDAC1, HSP90AA1, IRAK1, RIPK1, and RIPK2 all having descendants in the interactome, and targets of protein kinase inhibitors and HDAC inhibitors being particularly central (Supplementary Table 1). To analyze the most critical targets for the crosstalk between SARS-CoV-2 and aging, we repeated the analysis in the interactome obtained without taking aging into account (Supplementary Fig. 19c). Interestingly, while HDAC1 and HSP90AA1 continued to have a widespread effect, the number of genes downstream of RIPK1 changed drastically to just 1, suggesting that RIPK1 plays a critical role in the SARS-CoV-2 and aging cross-talk. In line with this, while the effect of HDAC inhibitors remained similar in the analysis without aging, the effect of protein kinase inhibitors changed drastically (Supplementary Table 1). Collectively, our combined analysis points to protein kinase inhibitors, and it in particular highlights RIPK1, a serine/ threonine–protein kinase, as one of the main targets against SARS-CoV-2 infections with a highly age-dependent role and the largest number of downstream differentially expressed genes in the combined SARS-CoV-2 and aging interactome.

## Discussion

The repurposing of drugs for SARS-CoV-2 has been a major challenge given the many pathways involved in host-pathogen interactions and the intricate interplay of SARS-CoV-2 with inflammatory pathways[13–16]. Interestingly, while both young and old individuals are susceptible to SARS-CoV-2 infection, the virus' pathogenicity is significantly more pronounced in the elderly[17,18]. Since the mechanical properties of the lung tissue change with aging[19], this led us to hypothesize an interplay between viral infection/replication and tissue aging[21], suggesting that this could play an important role in drug discovery programs. While ongoing drug repurposing efforts have analyzed host–pathogen interactions and the associated gene expression programs[6,23], they have lacked integration with aging. More generally, while a number of data-driven and experimental approaches have been proposed for drug identification and repurposing[1], a platform that systematically integrates different data modalities including transcriptomic, proteomic and structural data into a principled causal framework to predict the effect of different drugs has been missing.

By combining bulk RNA-seq data from GTEx[24] and Blanco et al.[23], we identified a critical group of genes that were differentially expressed by aging and by SARS-CoV-2 infection. While previous analysis relied primarily on contrasting the expression in cells with and without SARS-CoV-2 infection[32], we made an attempt to separate the effect of the ACE2 receptor alone and the effect of SARS-CoV-2 in cells without ACE2 receptor to extract a more refined differential expression pattern of ACE2-mediated SARS-CoV-2 infection. While previous computational efforts to repurpose drugs have mainly considered two approaches: (1) identifying drug targets by analyzing disease networks based for example on PPI or transcriptomic data[4–6], and (2) identifying

drugs by matching their signature (e.g., obtained from the CMap project[2]) to the reverse disease signature[3], we developed a principled causal framework that encompasses these two approaches. First, in order to ensure that the CMap database, which measures expression using 1000 representative genes, would be useful in the context of SARS-CoV-2, we validated that the intersection of these genes with the SARS-CoV-2 differentially expressed genes was significant. Second, to establish drug signatures based on the CMap database, we employed a particular autoencoder framework[41]. Rather unintuitively, we showed that using an overparameterized autoencoder, i.e., by using an autoencoder not to perform dimension reduction as usual but to instead embed the data into a higher-dimensional space, aligned the drug signatures across different cell types. This allowed constructing synthetic interventions, i.e., to predict the effect of a drug on a cell type without measuring it by using other cell types to infer it. Third, to identify drug targets in the pathways intersecting SARS-CoV-2 and aging, we connected the differentially expressed genes in the PPI network using a Steiner tree analysis[30] and intersected the resulting interactome with high-affinity targets of the drugs obtained using the overparameterized autoencoder framework. Finally, while computational drug discovery programs have been largely correlative[1], we made use of recent causal structure discovery algorithms[11,47,48] to validate the identified drug targets and their downstream effects, thereby identifying protein kinase inhibitors such as axitinib, dasatinib, pazopanib, and sunitinib as drugs of particular interest for the repurposing against COVID-19. Among the various protein kinases, in particular from the family of serine/threonine–protein kinases, identified by our drug repurposing pipeline, RIPK1 was singled out by our causal analysis as being upstream of the largest number of genes that were differentially expressed by SARS-CoV-2 infection and aging, while losing its central role in the corresponding gene regulatory network without taking aging into account. Notably, RIPK1 has been shown to bind to SARS-CoV-2 proteins[6] and has also been found to be in an age-dependent module[32]. RIPK1 belongs to an interesting family of proteins comprising of a kinase domain on the N terminus and a death domain on the C terminus; activation of the kinase domain has been associated with epithelial cell homeostasis, while activation of the death domain leads to triggering necroptotic or apoptotic pathways[50,51], the death pathways potentially triggering tissue fibrosis[52]. Interestingly, our differential expression analysis found RIPK1 to be upregulated with SARS-CoV-2 infection. We hypothesize that upon SARS-CoV-2 infection in older individuals the death pathways may be favored, thereby leading to fibrosis and increased blood clotting. Consistent with this, recent postmortem lung tissue biopsies of SARS-CoV-2 human patients revealed a fibrotic epithelium and increased blood clotting[53,54].

In order to test how specific our findings are to SARS-CoV-2 and demonstrate the broad applicability of our pipeline, we repeated the analysis on gene expression data available from[23] for the respiratory syncytial virus (RSV) and influenza A virus (IAV); see Supplementary Note for a detailed description of the analysis. Differential gene expression analysis showed that the intersection of the identified genes with RSV and IAV was only 3.19% and 19.6%, respectively (Supplementary Fig. 20). Comparing the drug lists resulting from the overparameterized autoencoder analysis for IAV and RSV to SARS-CoV-2 shows that the drug rankings for SARS-CoV-2 and RSV are significantly different, while the rankings for SARS-CoV-2 and IAV are more similar, but less so than between different SARS-CoV-2 datasets (Supplementary Figs. 21 and 10). The Steiner tree analysis further reinforced these findings (Supplementary Fig. 22), which are in line with SARS-CoV-2 and IAV having more similar clinical symptoms with higher morbidity and fatality rates in the aging population, while RSV is riskier for young children.

Collectively, our results highlight the importance of RIPK1 in the interplay between SARS-CoV-2 infection and aging as a potential target for drug repurposing programs to be administered postinfection. There are various drugs currently approved that non-specifically target RIPK1 (such as pazopanib and sunitinib) as well as under investigation that are highly specific to RIPK1[55,56]. Given the distinct pathways elicited by RIPK1, there is a need to develop appropriate cell culture models that can differentiate between young and aging tissues to validate our findings experimentally and allow for highly specific and targeted drug discovery programs. While our method is broadly applicable, we note several limitations. First, our drug repurposing pipeline relies on the availability of RNA-seq data from normal and infected/diseased cells in the cell type of interest and therefore the availability of such data is necessary for the application of our platform. Second, since our autoencoder is trained on CMap data, which only contains the expression of 1000 genes (L1000 genes), it is possible that the signal of the infection may not be captured by these 1000 genes. However, this can be checked by assessing whether there is a statistically significant overlap between the L1000 genes and the differentially expressed genes in the disease/infection of interest, which we performed in our analysis for SARS-CoV-2. Finally, since the CMap data contains a limited set of drugs, it is possible that none of the drugs are anticorrelated with the disease signature, thus preventing the user from identifying drug candidates. While our work identified particular drugs and drug targets in the context of COVID-19, our computational platform is applicable well beyond SARS-CoV-2, and we believe that the integration of transcriptional, proteomic, and structural data with network models into a causal framework is an important addition to current drug discovery pipelines.

## Methods

**Bulk gene expression data**. The RNA-seq gene expression data related to SARS-CoV-2 infection in A549 and A549-ACE2 cells were obtained from ref. [23] under accession code GSE147507. The RNA-seq data of lung tissues for the aging analysis was downloaded from the GTEx Portal (https://gtexportal.org/home/index.html) along with metadata containing the age of the individual from whom the RNA-seq sample was obtained. The RNA-seq raw read counts were transformed into quantile normalized, $\log_2(x+1)$ scaled RPKM values, following the normalization performed in ref. [2].

**Differential expression analysis**. For differential expression analysis, we focused on genes that were highly expressed, filtering out any genes with $\log_2$ (RPKM +1) < 1 for all considered datasets. In order to determine the ACE2-mediated SARS-CoV-2 genes, we computed three different $\log_2$-fold changes based on the data from[23]. Namely, we defined as ACE2-mediated SARS-CoV-2 genes all genes that had an absolute $\log_2$-fold change between A549-ACE2 cells infected with SARS-CoV-2 and A549-ACE2 cells above the threshold, excluding genes that had an absolute $\log_2$-fold change above the same threshold in A549-ACE2 cells versus A549 cells and also excluding genes that had an absolute $\log_2$-fold change above the same threshold in A549 cells infected with SARS-CoV-2 versus normal A549 cells. In other words, the ACE2-mediated SARS-CoV-2 genes were defined as the genes denoted in red in the Venn diagram in Fig. 2b (with pink, brown, and yellow subsets removed). The absolute $\log_2$-fold change threshold was determined such that the number of ACE2-mediated SARS-CoV-2 genes was 10% of the protein-coding genes.

In order to determine the age-associated genes, we analyzed lung tissue samples obtained from the GTEx portal (https://gtexportal.org/home/index.html) from individuals of varying ages. We computed the absolute $\log_2$-fold change between samples of the lung tissue from older (70–79 years old) and younger (20–29 years old) individuals, defining the age-associated genes as the top 10% of protein-coding genes with the highest absolute $\log_2$-fold change. We also considered defining age-associated genes based on the absolute $\log_2$-fold change comparing individuals who are 20–29 years old versus 60–79 years old, which yielded similar age-associated genes, with 1339 out of the 1923 genes in common between the two sets as shown in Supplementary Fig. 3b.

**Gene ontology enrichment analysis**. Gene ontology analysis was performed on a given gene set using GSEApy (v0.9.18), keeping the top ten gene ontology biological process terms with the lowest $p$ values. All reported terms had $p$ values ≤ 0.05, after adjusting for multiple hypothesis testing using the Benjamini–Hochberg procedure.

**L1000 gene expression data from CMap**. The CMap data measured via L1000 high-throughput reduced representation expression profiling, which quantifies the expression of 1000 landmark genes, was obtained from[2] under accession code GSE92742. We chose level 2 data, truncated to only the genes that were also measured by ref. [23], and then performed $\log_2(x+1)$ scaling and min–max scaling on each of the resulting 911-dimensional expression vectors.

**Combined autoencoder and synthetic interventions framework**. We first describe our training procedures for the autoencoder framework. CMap contains a total of 1,269,922 gene expression vectors and we performed a 90-10 training-test split resulting in 1,142,929 training examples and 126,993 test examples. We selected the best model by applying early stopping with an upper bound on the number of total epochs being 150. Note that this is well past the usual early stopping method of applying a patience strategy with the patience of at most ten epochs[57]. All hyperparameter settings, optimizer details, and architecture details are presented in Supplementary Fig. 6c. To summarize, we considered a range of fully connected autoencoders with varying width and nonlinearity, and we used Adam with a learning rate of $1^{-4}$ for optimization. To compute the drug signatures via the trained autoencoder, we used as embeddings the output of the first hidden layer prior to application of the activation function.

Drug signatures for the A549 cells (and similarly for the MCF7 and HCC515 cells) in CMap were computed by taking the difference between the mean embedding for the A549 samples with drug and the mean embedding for the A549 control (DMSO) samples. To remove batch effects, we performed $k$-means clustering of the control samples in the embedding space and removed all points falling in the smaller of the two clusters (see Supplementary Fig. 5b). Subsequent analysis of the removed cluster revealed that it consisted of samples with a minimum gene-expression value of 1 (after $\log_2(x+1)$ scaling), while all other gene expression values fell in the range of [5, 13], thereby providing further reason for the removal of this cluster. Next, we briefly describe the framework of the synthetic intervention and how the embedding from our trained overparameterized autoencoder is used for this. The traditional application of synthetic interventions[27,28] in the context of drug repurposing would proceed as follows: when a drug signature is unavailable on a given cell type but is available on other cell types, we would express the cell type as a linear combination of the other cell types and use this linear combination to predict the signature on the cell type for which data is unavailable. Since we demonstrated that overparameterized autoencoders align drug signatures between different cell types (Supplementary Fig. 8), instead of using a linear combination of drug signatures across cell types, we can simply use one of the available drug signatures as the synthetic intervention. In particular, in this work, we used drug signatures on MCF7 cells to construct synthetic interventions for A549 cells. We also considered drug signatures on HCC515 cells; however, there was only one FDA-approved drug that was applied to HCC515 cells which was not also applied to A549 cells in CMap. While this analysis did not help to increase the number of considered drugs, we used the data on HCC515 cells in conjunction with the data on A549 and MCF7 cells to validate that the overparameterized autoencoder aligns the signatures of drugs between different cell types (Fig. 3d and Supplementary Fig. 8).

**Cosine similarity between perturbations**. For each cell type and perturbation, we computed a cell type-specific "perturbation signature", which is defined as the difference between the average gene expression of a cell type under that perturbation and under the control perturbation, DMSO. Then, for each perturbation, we computed the cosine similarity ($\frac{\mathbf{a}\cdot\mathbf{b}}{\|\mathbf{a}\|\|\mathbf{b}\|}$) between the perturbation vectors for all pairs of cell types which received that perturbation in CMap. For example, daunorubicin was applied to 14 cell types in CMap, resulting in $\binom{14}{2} = 91$ cosine similarities associated with daunorubicin. All cosine similarities were plotted (Fig. 3e).

**Steiner tree analysis**

*Human PPI network*. A weighted version of the publicly available IRefIndex v14 (IREF) human PPI network[42] was retrieved from the OmicsIntegrator2 GitHub repository (http://github.com/fraenkel-lab/OmicsIntegrator2). The interactome contains 182,002 interactions between 15,759 proteins. Each interaction $e$ has an associated cost $c(e) = 1 - m(e)$ where the score $m(e)$ is obtained using the MIScore algorithm[58], which quantifies confidence in the interaction $e$ based on several evidence criteria (e.g., number of publications reporting the interaction and corresponding detection methods).

*Human-SARS-CoV-2 PPI network*. A high-confidence host–pathogen interaction map of 27 SARS-CoV-2 viral proteins with HEK293T proteins[6] was retrieved from NDEx, which reports interactions with 332 human proteins.

*Drug–target interaction data*. Data on the targets of drugs was obtained from DrugCentral, an online drug information resource, which includes drug–target interaction data extracted from the literature along with metrics (such as inhibition constant $K_i$, dissociation constant $K_d$, effective concentration EC50, and inhibitory concentration IC50) measuring the affinity of the drug for its target[44,45]. Drugs in the database are approved by the FDA and may also be approved by other regulatory agencies (such as the EMA). From this database, we filtered out compounds targeting non-human proteins. We also discarded drug–target pairs with affinity metrics ($K_i$, $K_d$, EC50, or IC50) higher than 10 μM, a commonly used threshold in the field. Based on this filtering we obtained a data set containing 12,949 high-affinity drug-target pairs involving 1457 unique human protein targets and 2095 unique compounds. This dataset was further restricted to drugs predicted to reverse the SARS-CoV-2 signature (correlation greater than 0.86 in the overparameterized autoencoder embedding). This correlation threshold was chosen to be the point at which the proportion of selected drugs decreases the most rapidly (Supplementary Fig. 23). As a result, the final drug–target data set included information on 2296 drug–target pairs involving 652 unique human gene targets and 117 unique FDA-approved drugs.

*Prize-collecting Steiner forest algorithm*. The Prize-Collecting Steiner Forest (PCSF) problem is an extension of the classical Steiner tree problem: Given a connected undirected network with non-negative edge weights (costs) and a subset of nodes, the terminals, find a subnetwork of minimum weight that contains all terminals. The resulting subnetwork is always a tree, which in general contains more nodes than the terminals; these are known as Steiner nodes. In the special case when there are only 2 terminals, this boils down to finding the shortest path between these nodes. The Steiner tree problem, in general, is known to be NP-complete, but various approximations are available. The PCSF problem generalizes this problem by introducing prices for the terminals (in addition to the edge costs already present in the Steiner tree problem) and a dummy node connected to all terminals. The problem is then to find a connected subnetwork that minimizes an objective function involving the cost of selected edges and the prizes of terminals that are missing from the subnetwork as detailed below; we used OmicsIntegrator2 to solve this optimization problem[30].

To formally introduce the objective function, let $G = (V, E, c(\cdot), p(\cdot))$ denote the undirected PPI network with protein set $V$ (containing $N$ proteins), interaction set $E$, edge cost function $c(\cdot)$, set of terminals $S \subset V$ (containing $N$ proteins) and attributed prizes $p(\cdot)$. The version of the PCSF problem solved by OmicsIntegrator2[30] and used in this article consists of finding a connected subnetwork $T = (V_T, E_T)$ of the modified graph $G^* = (V \cup \{r\}, E \cup \{\{r,s\}:s \in S\})$ that minimizes the objective function

$$\psi(T) = b \sum_{v \notin V_T} p(v) + \sum_{e \in E_T} c^*(e) \tag{1}$$

The node $r$ is a dummy root node connecting all terminals in the network. The parameter $b \in \mathbb{R}^+$ linearly scales the node prizes (which are non-zero for terminal nodes exclusively), and the modified edge cost function $c^*(\cdot)$ can be expressed as follows. For any edge $e = \{x, y\}$

$$c^*(e) = \begin{cases} c(e) + \frac{d_x d_y}{d_x d_y + (N - d_x - 1)(N - d_y - 1)} 10^g & \text{if } e \in \mathrm{E} \\ w & \text{if } e \in \{\{r,s\} : s \in \mathrm{S}\} \end{cases} \tag{2}$$

where $d_x$ denotes the degree of node $x$ in $G$ and $g, w \in \mathbb{R}^+$ are tuning parameters. If the resulting tree contains the root node $r$, $r$ is removed from the tree, and the output is an ensemble of trees, a forest. The final output, the interactome, is the subnetwork in the PPI network induced by the nodes of this forest.

*Selection of terminal nodes*. Results from the differential expression analysis yielded 219 protein-coding genes that were associated with both aging and SARS-CoV-2 infection. Of particular interest among these genes were 181 genes that showed concordant regulation, i.e., they were either upregulated in both SARS-CoV-2 infection and aging or downregulated in both SARS-CoV-2 infection and aging. Intersecting the proteins corresponding to these 181 genes with proteins in the IREF interactome resulted in 162 proteins. These 162 proteins were selected as terminal nodes for the PCSF algorithm and prized according to their absolute $\log_2$-fold change between SARS-CoV-2-infected A549-ACE2 cells and normal A549-ACE2 cells (Supplementary Fig. 11).

*Parameter sensitivity analysis*. Running the PCSF algorithm in the OmicsIntegrator2 required specifying three tuning parameters: $g$, $w$, and $b$. In order to guarantee the robustness of the resulting network with respect to moderate changes in these parameters, we selected the parameters based on a sensitivity analysis.

The parameter $g$ modifies the background PPI network by imposing an additive penalty on each edge based on the degrees of the corresponding vertices. It reduces the propensity of the algorithm to select hub nodes connecting many proteins in the interactome. While this feature may be relevant in certain biological

applications, it was not necessarily the case in our work since high degree nodes may be of interest for the purpose of drug target identification. In the cost function in Eq. (2), the absence of penalty corresponds to $g = -\infty$. However, the OmicsIntegrator2 implementation only allows for $g \in \mathbb{R}^+$. In Supplementary Fig. 12a1, we reported boxplots of penalized edge costs in the IREF interactome for different values of $g$. These boxplots suggest that the hub penalty parameter $g = 0$ yields similar edge costs to the desired setting where $g = \infty$. For this reason, we chose the value $g = 0$ in all OmicsIntegrator2 runs in this work.

The parameter $w$ corresponds to the cost of edges connecting terminal nodes to the dummy root $r$. This parameter influences the number of trees in the Steiner forest. If $w$ is chosen too low compared to the typical shortest path cost between two terminals, a trivial solution will connect all terminal nodes via $r$, leading to fully isolated terminals in the final forest. For high values of $w$ the PCSF algorithm will not include the root $r$ and output a connected network. Based on the histogram of the cost of the shortest path between any two terminals in the IREF interactome reported in Supplementary Fig. 12a2, we ran a sensitivity analysis for $w$ in the range [0.2, 2].

The parameter $b$ linearly inflates the prizes of terminal nodes in the objective function. Higher values of $b$ result in more terminal nodes in the final PCSF. We analyzed edge costs in the network to determine a suitable range for $b$ so as to include many terminal nodes in the resulting interactome. Supplementary Fig. 12a1 shows that the maximum edge cost in the network for $g = 0$ was lower than 1, which meant that making $b$ of order greater than 1 was necessary to ensure that trading off cost of edges added and prizes collected in the solution would rarely require discarding a terminal node. For this reason, we ran a sensitivity analysis for $b$ in the range [5, 50].

Based on the previous considerations we fixed $g = 0$ and ran a sensitivity analysis as described in Supplementary Fig. 12b with $w \in \{0.2, 0.4, 0.6, 0.8, 1, 1.2, 1.4, 1.6, 1.8, 2\}$ and $b \in \{5, 10, 15, 20, 25, 30, 35, 40, 45, 50\}$. We obtained 100 PCSFs, each corresponding to a particular choice of $(w, b)$. All of them included the entire terminal set $S$, the desired property resulting from the chosen range of the values of $b$. To analyze the robustness of the resulting networks to changes in the parameters, we analyzed the matrix $M \in [0, 1]^{100 \times 100}$ defined by

$$M_{ij} = \frac{\left| \left\{ \begin{array}{c} \text{nodes in} \\ \text{network } i \end{array} \right\} \cap \left\{ \begin{array}{c} \text{nodes in} \\ \text{network } j \end{array} \right\} \cap \mathcal{C} \right|}{\left| \left( \left\{ \begin{array}{c} \text{nodes in} \\ \text{network } i \end{array} \right\} \cup \left\{ \begin{array}{c} \text{nodes in} \\ \text{network } j \end{array} \right\} \right) \cap \mathcal{C} \right|} \qquad (3)$$

for every pair of PCSFs $i$ and $j$ corresponding to parameters $(w_i, b_i)$ and $(w_j, b_j)$, respectively. Supplementary Fig. 12c displays the heatmaps of this matrix. We considered three different node sets $\mathcal{C}$, namely the set of all nodes in the input PPI network (Supplementary Fig. 12c1), the subset of terminal nodes ($\mathcal{C} = S$, Supplementary Fig. 12c2), and the subset of SARS-CoV-2 interaction partners (Supplementary Fig. 12c3). Supplementary Fig. 12c1–c3 illustrate that choosing any $(w, b) \in [1.2, 2] \times [5, 50]$ led to the same connected PCSF with 252 nodes and 1003 edges. This network is robust to moderate parameter changes for $w$ and $b$. Collectively, this sensitivity analysis motivated the choice of $g = 0$, $w = 1.4$, and $b = 40$ used to obtain the interactome in Fig. 4b, where nodes are grouped by general function. The same interactome is presented in Supplementary Fig. 13 with nodes grouped by the general process. Note that since this interactome included all terminals and did not include the root node, it is equivalent to the solution of the classical Steiner tree problem.

*Neighborhood analysis.* For the interactomes obtained in this work, we reported two-nearest-neighborhoods of genes of interest in Fig. 4c for the interactome of Fig. 4b, in Supplementary Fig. 16 for the interactome of Supplementary Fig. 15, and in Supplementary Fig. 17d for the interactome in Supplementary Fig. 17c. Depending on the interactome, genes of interest include SARS-CoV-2 interaction partners (e.g., EXOSC5, FOXRED2, LOX, RBX1, and RIPK1) as well as genes of potential therapeutic interest (e.g., HDAC1, EGFR). Neighborhood plots were enriched with information such as SARS-CoV-2 interaction partners and FDA-approved, high affinity (based on data from DrugCentral) drugs with high correlation to the reverse SARS-CoV-2 infection signature. To improve the legibility of the neighborhood networks, we discarded the highly connected hub node UBC (connected to 62% of proteins in the IREF network). To further improve legibility, we applied an upper threshold on edge cost (i.e., only visualizing high confidence edges) when the neighborhood networks were too densely connected. We generally chose this threshold at 0.53, with the exception of the LOX neighborhood (0.58) and the FOXRED2, ETFA, and GNB1 neighborhoods (no thresholding). For each edge $e$ in a given neighborhood, we defined the min–max scaled edge confidence $C(e)$ as

$$C(e) = \frac{\max\limits_{e' \in \mathcal{E}} c(e') - c(e)}{\max\limits_{e' \in \mathcal{E}} c(e') - \min\limits_{e' \in \mathcal{E}} c(e')} \in [0, 1] \qquad (4)$$

where $\mathcal{E}$ denotes the edge set of the corresponding interactome and $c(e)$ denotes the cost of edge $e$ in the PPI network. This confidence metric was used to color edges in the neighborhood plots.

*Addition of SARS-CoV-2 interaction partners to the terminal node list.* In order to understand which other SARS-CoV-2 protein interaction partners were in the neighborhood of the identified interactome, we also ran the PCSF algorithm on the IREF PPI network using the SARS-CoV-2 and aging terminal list augmented with all known SARS-CoV-2 interaction partners. All SARS-CoV-2 interaction partners (with the exception of EXOSC5, FOXRED2, and LOX which were already present in the original terminal gene list) were given a small prize $p$. This prize was chosen by sensitivity analysis over a range of possible values from $p = 0$ (5 SARS-CoV-2 interaction partners initially selected by the method: EXOSC5, FOXRED2, LOX, RBXL1, and RIPK1) to $p = 0.02$, beyond which all 332 known SARS-CoV-2 interaction partners belonged to the computed interactome. The fine-grained analysis revealed that choosing $p \in [4 \times 10^{-4}, 10^{-3}]$ leads to interactomes which include a stable set of 7 SARS-CoV-2 interaction partners, the five present initially plus CUL2 and HDAC2 (Supplementary Fig. 14a). Supplementary Fig. 14b, c display heatmaps of the matrix $M \in [0, 1]^{16 \times 16}$ defined as

$$M_{ij} = \frac{\left| \left( \left\{ \begin{array}{c} \text{nodes in} \\ \text{network } i \end{array} \right\} \setminus \left\{ \begin{array}{c} \text{nodes in} \\ \text{network } j \end{array} \right\} \right) \cap \mathcal{C} \right|}{\left| \left\{ \begin{array}{c} \text{nodes in} \\ \text{network } i \end{array} \right\} \cap \mathcal{C} \right|} \qquad (5)$$

for every pair of PCSFs $i$ and $j$ corresponding to parameters $p_i$ and $p_j$, respectively. For the sensitivity analysis, we considered two different node sets $\mathcal{C}$, namely the set of all nodes in the input PPI network (Supplementary Fig. 14b) as well as the subset of SARS-CoV-2 interaction partners (Supplementary Fig. 14c). Supplementary Fig. 14b shows that the obtained interactome was stable over the range $p \in [7 \times 10^{-4}, 10^{-3}]$. Supplementary Fig. 14c shows that all SARS-CoV-2 interaction partners collected in the interactome when $p \in [7 \times 10^{-4}, 10^{-3}]$ were also collected for higher values of $p$, which is a consequence of the observation from Supplementary Fig. 14b. We used the value $p = 8 \times 10^{-4}$ for all subsequent analyses and figures, including Supplementary Fig. 15 and Supplementary Fig. 16.

*Randomization and robustness analysis.* We conducted several randomization assessments to understand the importance of each step in the pipeline, analyzing the impact of changes in the RNA-seq expression data, the underlying PPI network, the CMap drug signatures, as well as the list of terminal genes on the final selection of drug targets and corresponding drugs. This was quantified by the frequency of appearance of each drug in the final drug list after 1000 randomization runs, for both drugs that were and that were not selected in the original non-randomized analysis. Results from this analysis suggest that the choice of terminal genes is the most critical step of the Steiner tree procedure; see Supplementary Note and Supplementary Table 2.

To ensure the robustness of our results to different ways of mitigating batch effects in the CMap dataset, we repeated the analysis by dropping all genes for which there was at least one sample containing a 1 in the expression value (reducing the total number of genes from 911 to 867 for the A549 cell line). As with the original batch correction approach, the resulting drugs consist mainly of protein kinase inhibitors (7 out of 9) and the drug targets are highly overlapping with the drug targets obtained from the original analysis (Supplementary Fig. 24).

**Single-cell RNA-seq analysis.** Single-cell RNA-seq for A549 cells was obtained from GSE81861[46], where each entry in the matrix represents the gene expression (FPKM) of gene $i$ in cell $j$. We preprocessed the data, keeping only genes that had a nonzero gene expression value in more than 10% of the cells, followed by the transformation of the data. Single-cell RNA-seq data for AT2 cells were obtained from http://www.nupulmonary.org/resources associated with ref. [49]. In order to avoid batch effects, we subset the data to include cells only from Donor 7 since that donor had the largest number of AT2 cells collected (4002 cells). We preprocessed the data using the same threshold as for A549 cells for filtering out genes across cells. Since single-cell RNA-seq data for AT2 cells were not yet normalized, we normalized the expression values across genes for each cell by the total RNA count for that cell, followed by $\log_2(x + 1)$ transformation of the data as for A549 cells.

**Evaluation of causal structure discovery algorithms.** Prior to reporting the results of learning gene regulatory networks on A549 and AT2 cells, we benchmarked several causal structure discovery methods on the task of predicting the effects of interventions using gene knockout and overexpression data collected on A549 cells as part of the CMap project[2], similar to prior evaluations of causal methods[11,12]. We estimated the gene regulatory network underlying the identified interactome in A549 cells using the prominent causal structure discovery methods PC, GES, and GSP[8,47,48]. Since not all edge directions are identifiable from purely observational data, these methods output a causal graph containing both directed and undirected edges. Since the advantage of causal networks is their ability to predict the effects of interventions on downstream genes, we evaluated these methods using interventions collected in CMap. In the following, we first describe how we estimated the effects of interventions based on the CMap data to use as ground truth for evaluating causal structure discovery methods. We focused our evaluation on genes and interventions that are shared between the combined SARS-CoV-2 and aging interactome and CMap knockout and overexpression experiments, resulting in 32 genes and 41 interventions (note that the number of

interventions is larger than the number of genes since in CMap interventions have been performed on genes that are not part of the L1000 landmark genes but are contained in the interactome). We formed a matrix of genes by interventions, where each $(i, j)$-entry in the matrix represents the $\log_2$-fold change in expression of gene $i$ when gene $j$ was intervened on in comparison to the expression of gene $i$ without intervention. We denoted by $Q$ the binary matrix of intervention effects with $Q_{ij} = 1$ if the sign of the $\log_2$-fold change for the $(i, j)$ entry was opposite for knockout and overexpression interventions to filter out unsuccessful interventions, the rationale being that knockout and overexpression should have opposite downstream effects. Thus $Q_{ij} = 1$ denotes that perturbing gene $j$ affects gene $i$ and hence that gene $i$ is downstream of gene $j$ (Supplementary Fig. 18a). Taking this matrix of interventional effects, $Q$, as the ground truth, we estimated the causal graph using the PC, GES, and GSP algorithms and determined the corresponding ROC curve, counting and edge from $j \rightarrow i$ as a true positive if $Q_{ij} = 1$ and a false positive otherwise (Supplementary Fig. 18b). In order to statistically evaluate whether the different algorithms performed better than random guessing, we sampled causal graphs (from an Erdös–Renyi model, where the edges were directed based on a uniformly sampled permutation) with different edge probabilities from the PPI network and calculated the corresponding number of true and false positives. For each false positive level, we created a distribution over true positives based on the sampled random causal graphs and calculated the $p$ value for the number of true positives obtained from the PC, GES, and GSP algorithms. We combined the $p$ values across different numbers of false positives using Fisher's method and used this combined $p$ value for evaluating whether the PC, GES, and GSP algorithms were significantly different from random guessing.

**Causal structure discovery for learning gene regulatory networks**. In order to learn the gene regulatory networks governing A549 and AT2 cells, we used the recent structure discovery method GSP[11,12,47] on single-cell RNA-seq data from A549 cells as well as AT2 cells with the PPI network on 252 nodes as a prior. We used GSP since based on the previous analysis it outperformed the PC and GES algorithms in terms of ROC analysis on predicting the effect of gene knockout and overexpression experiments in A549 cells ($p$ value = 0.0177 for GSP, $p$ value = 0.0694 for GSP and $p$ value = 0.5867 for GES); in addition, GSP is also preferable from a theoretical standpoint, since it is consistent under strictly weaker assumptions than the PC and GES algorithms[47]. To obtain an estimate of the causal graph that is robust across hyperparameters and data subsampling, we used stability selection[59]. In short, stability selection estimates the probability of selection of each edge by running GSP on subsamples of the data. Aggregating selection probabilities across algorithm hyperparameters (in this case the $\alpha$-level for conditional independence testing), edges with high selection probability (0.3 for A549 cells and 0.4 for AT2 cells) were retained. The threshold for AT2 cells was chosen so as to approximately match the number of edges in the A549 network.

**Reporting summary**. Further information on research design is available in the Nature Research Reporting Summary linked to this article.

## Data availability
All datasets used in this work are publicly available from the following sources: The gene expression data for SARS-CoV-2 was obtained from GSE147507[23] and the gene expression data for the aging analysis was obtained from https://gtexportal.org/home/index.html[24]. The CMap data was downloaded using accession code GSE92742[2]. We used the PPI network from http://github.com/fraenkel-lab/OmicsIntegrator2 (IRefIndex Version 14)[42] and drug–target data from DrugCentral[44,45]. The single-cell RNA-seq data for the causal analysis was obtained from GSE81861[46] for A549 cells and http://www.nupulmonary.org/resources for AT2 cells associated with[49]. The host–pathogen interactions of SARS-CoV-2 proteins were obtained from http://www.ndexbio.org/#/network/5d97a04a-6fab-11ea-bfdc-0ac135e8bacf[6].

## Code availability
We relied on open source libraries to build our analysis pipeline. In particular, we used R (v3.6) package pcalg (v2.6) and the following python (3.7) packages: OmicsIntegrator2 (v2), causaldag (v0.1a133), GSEApy (v0.9.18), networkx (v2.4), numpy (v1.17.3), pandas (v0.25.3), PyTorch (v1.6), scikit-learn (v0.22.2), scipy (v1.4.1), cmapPy (v4.0.1), and graphviz (v2.40.1). Our code is available at https://github.com/uhlerlab/covid19_repurposing[60].

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

## Acknowledgements

A.B. was supported by J-WAFS and J-Clinic for Machine Learning and Health at MIT. A.R. was supported by the National Science Foundation (DMS-1651995) and IBM. C.S. and K.D.Y. were supported by the National Science Foundation (NSF) Graduate Research Fellowships and ONR (N00014-17-1-2147 and N00014-18-1-2765). G.V.S. was supported by ETH funding. C.U. was partially supported by NSF (DMS-1651995), ONR (N00014-17-1-2147 and N00014-18-1-2765), IBM, and a Simons Investigator Award. The Titan Xp used for this research was donated by the NVIDIA Corporation.

## Author contributions

All authors designed the research. A.B., L.C., A.R., C.S., and K.D.Y. developed and implemented the algorithms and performed model and data analysis. A.B., L.C., A.R., G.V.S., and C.U. wrote the paper.

## Competing interests

The authors declare no competing interests.
