## [Peer Review File · Nature Communications]

Reviewers' Comments:

Reviewer #1:

Remarks to the Author:

Belyaeva et al. propose a computational approach for repurposing drugs to treat COVID19. The platform integrates multi-omics data and multiple datasets to predict new candidate drugs and identify key gene targets associated with these targets, but in particular while considering age, an important factor in the severity and lethality of the disease. An initial concern was the specificity of these results to A549 cells which are tumor-derived, immortalized, and have many genomic and chromosomal abnormalities. These are work horse cells, but the lack of testing with other cell lines was concerning, though the authors did show that their autoencoder approach did show good representation across cell lines, so this concern was tempered by the results. The overall methodology is quite interesting and the results are also potentially useful. The identification of RIPK1 is also potentially interesting and could be explored further. However, there are some concerns that should be addressed. It is understood that the authors are limited to the data that has been generated and the timeliness of the study, so no experiments are requested, but a few computational experiments and some textual additions in the discussion would address many of the concerns listed below.

- I keep coming back to the same question, which is, are these results specific to SARS-CoV-2 or are these results generic to any viral infection? Is there anything unique to the biology that is discovered or the processes/pathways/genes found? In fact, the results don't necessarily need to be specific to SARS-CoV-2 to have a clinical impact, but it would be nice to know that these results are a bit more tuned to this coronavirus as opposed to any other virus. Could the authors find another dataset or two that could be used to show that the results are unique or which results are unique to SARS-CoV-2?
- The correlations listed in 3f are very close to each other. I have no sense of if a score of .879 and .883 are significantly different from each other. The authors should consider some kind of bootstrapping of the data to determine confidence intervals on the score to determine the robustness in rankings. It could be that a drug ranked 50th, for example, is statistically equivalent to the top ranked drug. Is there a way to give a better sense of how confident the drug predictions are vs. a hard to interpret correlation with the overparameterize autoencoder latent space?
- Also regarding the correlations in 3f, I don't have a sense of how different the rankings of the proposed method vs. Original or PCA. That is, if the rank ordering of the drugs by the proposed method are nearly identical to the rank ordering of the other methods, then, at least for prioritizing drugs, this approach does not offer any new insights.
- Overall, the paper is well written and clear, but I struggled a bit with the results given the full analysis. That is, what are the drugs really aimed to target. It would be nice to have a very clear statement in the intro and discussion about the eventual target of the drugs. The analysis was done with human data and considered aging and the analysis defines drugs that "reverse" the effect of SARS-CoV-2 infection, but in the context of an ACE inhibitor, so are the drugs targeting an infected patient or supposed to be used in a preventive setting. Are they only targeting lung tissue? And only in older people? I would like a bit more context of the ultimate purpose of the drugs given the results in the context of the data and analysis applied.
- Please add a limitations or caveats paragraph in the discussion to highlight a few of the limitations of this study. While the authors did show that the autoencoder can be applied to other cell lines, the study is fully based on the A549 cells, which is an inherent biological limitation. The other items mentioned above could be added to this paragraph as well.

Minor Points

- Figure 2f, please label the columns
- Figure 2g, add individual ranks as columns to see what the ranks are per experiment
- Figure 3a, not needed, but it might help to show the genes that are L1000 and DE in a different color to give a sense to the number of genes that fall into both categories.
- Figure 4d is not readable with very tiny text, same with the gene names in the network figures.

Reviewer #2:

Remarks to the Author:

The paper "Causal Network Models of SARS-CoV-2 Expression and Aging to Identify Candidates for Drug Repurposing" presents a computational systems medicine approach to drug target prediction for COVID-19. Essentially, the computational pipeline integrates transcriptomics, proteomics and structure data as well as drug screening data with biomolecular interaction networks. Interestingly, the authors performed the analyses such that aging-specific profiles are taken into account. The result is a candidate mechanism presumably highlighting serine/threonine as well as tyrosine kinases as targets for inhibition.

Essentially, the authors identify genes differentially expressed in aging as well as SARS-CoV-2 infected cells. Through CMAP embedding, approved drugs are prioritized across different cell types and generalized for SARS-CoV-2. Finally, the authors run a Steiner tree analysis to extract age-specific and infection-specific PPI subnetworks, which are then intersected to extract a candidate mechanism, which is potentially enriched with druggable target genes driving age-dependent COVID-19 disease progression.

Along the way many different computational tools are employed, all of which have different preconditions, advantages and (most importantly) require meta-parameters and cutoffs to be set. While a big part of the paper deals with justifications for each of the steps, one might have the impression that different tools and settings have been employed until a desired result was observed.

The paper in my opinion either needs (1) a preclinical / wet lab follow-up to demonstrate that the findings have clinically exploitable impact, (2) evaluation on completely independent data of different type(s) to backup the results/conclusions, or (3) an interactive software or web tool allowing other researchers to replicate the results and to visually explore the results for different algorithms and parameter choices - allowing others to bring in own background knowledge and to judge and modify the intermediate results properly.

The presented work - although showing significance of the individual steps - at the very least also requires a proper significance analysis of the overall strategy. If the exact same data is used and the exact same thresholds, parameters, and algorithms, how would the results change when running on (1) randomized PPI networks with similar graph properties, (2) permuted expression data, (3) randomly selected CMAP signatures, (4) randomly picked genes instead of the Steiner-tree-selected ones, etc. The strategy would still provide results, but (hopefully) very different ones. How different? Would the "performance" drop? And how would "performance" be measured?

Bottom line: The paper needs a higher real-world evidence level to demonstrate its impact as well as a computational robustness analysis of the overall strategy.

Reviewer #3:

Remarks to the Author:

The authors developed a platform that systematically integrates available transcriptomic, proteomic and structural data to identify robust druggable protein targets. Their results highlight the importance of RIPK1 in the interplay between SARS-CoV-2 infection and aging as a potential target for drug repurposing programs. The methods and results reported are interesting especially that they integrate aging signatures into drug discovery platforms, which is very novel. I have several major concerns about this work.

- 1) Though "Rigorous in vitro experiments as well as clinical trials are needed to validate the identified candidate drugs" is declared. I think the authors need to do this validation themselves (or by cooperation) to make a full story.
- 2) It seems that a pipeline of methods and datasets are used and integrated. Are there any novelty in the algorithms? I cannot figure out if any new algorithms are developed or customized to problems studied. Though figure 1 shows the overview of computational drug repurposing platform for COVID-19. I suggest a schematic diagram for the computational methods and how they were connected is required.
- 3) The authors claimed that the drug discovery platform is broadly applicable. Can they give a minimal requirement for the input data to the users? I mean more discussion on the methods and algorithms is needed.
- 4) It is really interesting that the importance of RIPK1 in the interplay between SARS-CoV-2 infection and aging as a potential target for drug repurposing programs was found. Can the authors give more discussion on and insight into what is the difference between young and old individuals after SARS-CoV-2 infection besides result representation.

Reviewer #1 response:

Belyaeva et al. propose a computational approach for repurposing drugs to treat COVID19. The platform integrates multi-omics data and multiple datasets to predict new candidate drugs and identify key gene targets associated with these targets, but in particular while considering age, an important factor in the severity and lethality of the disease. An initial concern was the specificity of these results to A549 cells which are tumor-derived, immortalized, and have many genomic and chromosomal abnormalities. These are work horse cells, but the lack of testing with other cell lines was concerning, though the authors did show that their autoencoder approach did show good representation across cell lines, so this concern was tempered by the results. The overall methodology is quite interesting and the results are also potentially useful. The identification of RIPK1 is also potentially interesting and could be explored further. However, there are some concerns that should be addressed. It is understood that the authors are limited to the data that has been generated and the timeliness of the study, so no experiments are requested, but a few computational experiments and some textual additions in the discussion would address many of the concerns listed below.

We thank the reviewer for the thoughtful comments, which helped improve our manuscript. We provide a point-by-point response to each comment below.

- I keep coming back to the same question, which is, are these results specific to SARS-CoV-2 or are these results generic to any viral infection? Is there anything unique to the biology that is discovered or the processes/pathways/genes found? In fact, the results don't necessarily need to be specific to SARS-CoV-2 to have a clinical impact, but it would be nice to know that these results are a bit more tuned to this coronavirus as opposed to any other virus. Could the authors find another dataset or two that could be used to show that the results are unique or which results are unique to SARS-CoV-2?

To answer this question, we repeated the most critical steps of our computational pipeline to obtain results for two additional viral infections: respiratory syncytial virus (RSV) and influenza A virus (IAV). Gene expression data for this analysis was obtained from [1], as was already the data used for our original analysis on SARS-CoV-2. In the following, we provide the differences in gene expression of these viruses, differences in predicted drug lists and differences in predicted gene targets. To summarize before providing the details of the analysis, we find that (1) the majority of the differentially expressed genes are different between SARS-CoV-2, IAV and RSV, (2) the ranked drug lists given by the autoencoder are different between SARS-CoV-2, IAV and RSV (although the difference is much smaller between SARS-CoV-2 and IAV than between SARS-CoV-2 and RSV), (3) the filtered drugs after the PPI interactome analysis also contain protein kinase inhibitors for IAV but none of the predicted drugs for SARS-CoV-2 are shared with RSV, (4) the gene targets of the chosen drugs are different between SARS-CoV-2, IAV and RSV. Overall, we observed that SARS-CoV-2 expression and thus the resulting drugs and drug targets are more similar to IAV than to RSV, which is in line with SARS-CoV-2 and IAV having similar clinical symptoms with higher morbidity and fatality rates in the aging population, while RSV is riskier for young children.

Figure 1: (a) Venn diagram of overlap between differentially expressed genes in SARS-CoV-2, RSV and IAV infections. (b) Heatmap of log₂ fold change of differentially expressed genes shared by SARS-CoV-2, IAV and RSV (first 3 genes), SARS-CoV-2 and IAV (40 genes), and SARS-CoV-2 and RSV (last 4 genes).

In the following, we provide more details on the analysis. First, we analyzed the gene expression signature of RSV and IAV as compared to SARS-CoV-2. We obtained from [1] RNA-seq samples of normal, RSV-infected and IAV-infected A549 cells and repeated the differential expression analysis steps for these viral infections. Fig. 1a above (which is Supplementary Fig. S20a in the revised Supplementary Materials) shows the overlap between identified genes for each infection with 3.19% and 19.6% of genes specific to SARS-CoV-2 also shared with RSV and IAV, respectively. Fig. 1b above (which is Supplementary Fig. S20b in the revised Supplementary Materials) shows the \log_2 fold change for genes shared between SARS-CoV-2, IAV and/or RSV.

Next, we analyzed the drug lists (drugs ranked by their inner product with the reverse disease signature) obtained by applying our over-parameterized autoencoder to the IAV and RSV expression data. In order to provide a quantitative comparison of the generated drug lists, we consider the curves shown in Fig. 2 and 3 below (which are Supplementary Fig. S21 and S10 in the revised Supplementary Materials). These curves are similar to receiver operating characteristic (ROC) curves, namely: given two drug lists with n drugs each, we consider the top k drugs and compute the number of drugs in common among these top k drugs for $k = 1, 2, \dots, n$. Thus, the x -coordinate in each plot indicates the proportion, k/n , of each drug list we considered and the y -coordinate is the size of the intersection of the two subsets normalized by k . The area under the curve (AUC) is a measure of similarity between drug lists. When two drug lists are exactly the same, the AUC is 1. On the other hand, it is important to note that as shown in Fig. 6a below, when the two drug lists are maximally different (i.e., one drug list is the reverse of the other), the AUC is $1 - \ln(2) \approx .306$. This is due to the fact that once k reaches $n/2$, the drug

Figure 2: Quantitative analysis of similarity between drug lists obtained using the overparameterized autoencoder on gene expression data from different virus infections. Comparison of drug lists from SARS-CoV-2 infected A549-ACE2 cells versus (a) RSV infected A549 cells, and (b) IAV infected A549 cells.

Figure 3: Quantitative analysis of similarity between drug lists obtained using the overparameterized autoencoder on gene expression data from different MOIs for A549 cells with and without ACE2 supplement. (a) Comparison of drug lists obtained from SARS-CoV-2 infected A549-ACE2 cells with MOI 2 and A549 cells with MOI 2, (b) A549-ACE2 cells with MOI 2 and A549-ACE2 cells with MOI 0.2, and (c) A549 cells with MOI 2 and A549-ACE2 cells with MOI 0.2. The similarity between the drug lists drops when comparing an MOI of 2 to an MOI of 0.2, which is consistent with the observation by [1] that low-MOI conditions did not stimulate an important interferon-I and -III response.

lists have to intersect. Fig. 2a-b show that that the drug lists for SARS-CoV-2 and RSV are significantly different and in fact very close to the lower bound, while the drug lists for SARS-CoV-2 and IAV are quite similar with an AUC of 0.843. To put this AUC into perspective and also analyze the robustness of the drug list obtained for SARS-CoV-2 (which was based on A549-ACE-2 cells with MOI of 2), we computed the drug lists also for SARS-CoV-2 infected A549 cells (without ACE-2 supplement) with an MOI of 2 and for A549-ACE2 cells with an MOI of 0.2; this gene expression data was also obtained from [1]. Fig. 3a-c show that the drug lists produced by considering other SARS-CoV-2 samples from [1] are very similar, indicating that the drug lists obtained by our analysis are robust and specific to SARS-CoV-2. The similarity between the drug lists is the highest (AUC= 0.946) when comparing SARS-CoV-2 infection in A549 cells with and without ACE-2 supplement, both infected with MOI of 2. The similarity between the drug lists drops when comparing an MOI of 2 to an MOI of 0.2, which is consistent with the observation by [1] that low-MOI conditions did not stimulate an important interferon-I and -III response and thus may result in different suggested drugs. However, these SARS-CoV-2 specific drug lists were still more similar (AUCs of 0.946, 0.875, 0.870) than the drug lists obtained from IAV (AUC of 0.843) or RSV (AUC of 0.308).

Figure 4: Drugs and their gene targets obtained from the prize-collecting Steiner tree analysis for IAV and RSV in comparison to our findings for SARS-CoV-2. (a) Venn diagram between selected drugs for IAV and SARS-CoV-2 using aging as a filter in the differential gene expression analysis for both viruses, and (b) Venn diagram for the respective gene targets. (c) Venn diagram between selected drugs for RSV and SARS-CoV-2 without taking aging into account for the differential expression analysis of RSV, and (d) Venn diagram for the respective gene targets. (e) Venn diagram between selected drugs for RSV and SARS-CoV-2 using aging as a filter in the differential gene expression analysis for both viruses, and (f) Venn diagram for the respective gene targets.

Next, we performed the Steiner tree analysis based on the identified differentially expressed genes for IAV and RSV as well as the drug lists obtained by the overparameterized autoencoder. As for SARS-CoV-2, since the morbidity and fatality rate of IAV is higher in the aging population, we computed a combined IAV and aging interactome. This consisted of 185 nodes and 486 edges based on 124 terminal genes. Since RSV is riskier in young children, but can also be serious for the aging population, we computed two interactomes, one without taking aging into account (234 nodes and 871 edges based on 139 terminal genes) and one combined with RSV and aging (303 nodes and 1177 edges based on 200 terminal genes) to make it more comparable to the other interactomes. To make the results comparable, since for SARS-CoV-2 we intersected the targets of the top 142 ranked drugs from the overparameterized autoencoder analysis with the interactome, we performed the analysis with the same number of drugs also for IAV and RSV. The resulting drugs and drug targets are shown in Fig. 4 above (which is Supplementary Fig. S22 in the revised Supplementary Materials). For IAV, this resulted in 20 drugs, 13 of which overlap with drugs identified in the SARS-CoV-2 analysis. These drugs target 9 proteins in the interactome, 2 of which are also present in the SARS-CoV-2 interactome, namely EGFR and RIPK1. For RSV with and without aging the resulting drug lists as well as their targets had no overlap with the ones identified by SARS-CoV-2. In particular, the identified drug lists contained no tyrosine kinase inhibitors, thereby indicating the specificity of our results to SARS-COV-2.

To reflect the above described analysis on IAV and RSV, we added the following text in the discussion section of the revised manuscript: "In order to test how specific our findings are to SARS-CoV-2 and demonstrate the broad applicability of our pipeline, we repeated the analysis on gene expression data available from [1] for respiratory syncytial virus (RSV) and influenza A virus (IAV); see Supplementary Note for a detailed description of the analysis. Differential gene expression analysis showed that the intersection of the identified genes with RSV and IAV was only 3.19% and 19.6%, respectively (Supplementary Fig. S20). Comparing the drug lists resulting from the overparameterized autoencoder analysis for IAV and RSV to SARS-CoV-2 shows that the drug rankings for SARS-CoV-2 and RSV are significantly different, while the rankings for SARS-CoV-2 and IAV are more similar, but less so than between different SARS-CoV-2 datasets (Supplementary Fig. S21 and S10). The Steiner tree analysis further enforced these findings (Supplementary Fig. S22), which is in line with SARS-CoV-2 and IAV having more similar clinical symptoms with higher morbidity and fatality rates in the aging population, while RSV is riskier for young children."

In addition, we added the following section on "Comparison of SARS-CoV-2 versus IAV and RSV" to the Supplementary Note:

"In order to test how specific our findings are to SARS-CoV-2 and demonstrate the broad applicability of our pipeline, we apply our computational pipeline to two additional viral infections: respiratory syncytial virus (RSV) and influenza A virus (IAV). As for SARS-CoV-2 infection, we obtain gene expression data for these viruses from [1]. First, we perform differential expression analysis for IAV and RSV (Supplementary Fig. S20) showing that only 3.19% and 19.6% of genes specific to SARS-CoV-2 are shared with RSV and IAV, respectively. Next, we apply our over-parameterized autoencoder and synthetic interventions framework to IAV and RSV to obtain drug lists ranked by their correlation with the reverse disease signature.

In order to quantitatively compare the drug lists obtained for RSV and IAV to the drug list for SARS-CoV-2, we measure the similarity of two rankings using curves akin to a receiver operating characteristic (ROC) curve, namely: given two rankings of n drugs, we consider the top k drugs in one of the lists and compute the number of drugs in common among these top k drugs for $k = 1, 2, \dots, n$. Thus, the x -coordinate in each plot indicates the proportion, k/n , of each drug list we consider and the y -coordinate is the size of the intersection of the two subsets normalized by k . The area under the curve (AUC) is a measure of similarity between two drug lists. When two drug lists are exactly the same, the AUC is 1 and when the two drug lists are maximally different (i.e., one drug list is the reverse of the other), the AUC is $1 - \ln(2) \approx .306$; see Supplementary Fig. S9a. Supplementary Fig. S21a-b show that that the drug lists for SARS-CoV-2 and RSV are significantly different and in fact very close to the lower bound, while the drug lists for SARS-CoV-2 and IAV are quite similar with an AUC of 0.843.

Finally, we perform the Steiner tree analysis based on the identified differentially expressed genes for IAV and RSV as well as the drug lists obtained by the overparameterized autoencoder. As for SARS-CoV-2, since the morbidity and fatality rate of IAV is higher in the aging population, we compute a combined IAV and aging interactome. This consists of 185 nodes and 486 edges based on 124 terminal

genes. Since RSV is riskier in young children, but can also be serious for the aging population, we compute two interactomes, one without taking aging into account (234 nodes and 871 edges based on 139 terminal genes) and one combined with RSV and aging (303 nodes and 1177 edges based on 200 terminal genes) to make it more comparable to the other interactomes. To make the results comparable, since for SARS-CoV-2 we intersected the targets of the top 142 ranked drugs from the overparameterized autoencoder analysis with the interactome, we perform the analysis with the same number of drugs also for IAV and RSV. The resulting drugs and drug targets are shown in Supplementary Fig. S22. For IAV, this results in 20 drugs, 13 of which overlap with drugs identified in the SARS-CoV-2 analysis. These drugs target 9 proteins in the interactome, 2 of which are also present in the SARS-CoV-2 interactome, namely EGFR and RIPK1. For RSV with and without aging the resulting drug lists as well as their targets have no overlap with the ones identified by SARS-CoV-2. In particular, the identified drug lists contain no tyrosine kinase inhibitors, thereby indicating the specificity of our results to SARS-CoV-2.”

- The correlations listed in 3f are very close to each other. I have no sense of if a score of .879 and .883 are significantly different from each other. The authors should consider some kind of bootstrapping of the data to determine confidence intervals on the score to determine the robustness in rankings. It could be that a drug ranked 50th, for example, is statistically equivalent to the top ranked drug. Is there a way to give a better sense of how confident the drug predictions are vs. a hard to interpret correlation with the overparameterize autoencoder latent space?

We thank the reviewer for this suggestion. We added Fig. 5 below (which is Supplementary Fig. S23 in the revised manuscript) to describe how we selected the threshold to identify candidate drugs based on the provided correlations. This figure shows the percentage of drugs (y-axis) with correlation higher than a given threshold (x-axis). The threshold chosen (0.86) corresponds to the largest jump in the y-values of the plot, which captures the largest change in the proportion of drugs for a small increase in correlation. To clarify this, we added the following sentence in the revised main text (Methods section): ”This correlation threshold was chosen to be the point at which the proportion of drugs decreases the most rapidly (Supplementary Fig. S23).”

- Also regarding the correlations in 3f, I don’t have a sense of how different the rankings of the proposed method vs. Original or PCA. That is, if the rank ordering of the drugs by the proposed method are nearly identical to the rank ordering of the other methods, then, at least for prioritizing drugs, this approach does not offer any new insights.

To answer this question, we compare the drug lists that we would obtain in the original space and in the PCA space to the ones obtained in the latent space using the same plot as in Fig. 3 above. This is shown in Fig. 6 (which is Supplementary Fig. S9 in the revised Supplementary Materials). While the drug lists based on the latent space embedding are similar to those based on an embedding in the original space (AUC= 0.904) and the PCA space (AUC= 0.901), note that these drug lists are more different

Figure 5: Selection of correlation threshold for identifying candidate drugs. Plot showing the percentage of drugs (y-axis) with correlation higher than a given threshold (x-axis). The vertical red line indicates the x-value (0.86) for which the y-value shows the largest jump and corresponds to the threshold used for the selection of drug candidates.

than the drug lists obtained when comparing A549-ACE2 versus A549 (AUC= 0.946; see Fig. 3a above). To reflect this analysis, we included the following text in the revised main text: "To compare the correlations obtained with the different embeddings, a list of the top ranked drugs is shown in Fig. 3f and the similarity between drug lists is quantitatively assessed by an analysis akin to a receiver operating characteristic (ROC) plot (Supplementary Note and Supplementary Fig. S9), showing that the drug lists obtained using an embedding in the PCA or the original space are similar but not identical to the autoencoder embedding (AUC of 0.901 and 0.904, respectively)....in fact the drug lists from A549 cells with and without ACE-2 supplement in the autoencoder embedding were more similar than the drug lists obtained from the PCA and the original space embedding."

- Overall, the paper is well written and clear, but I struggled a bit with the results given the full analysis. That is, what are the drugs really aimed to target. It would be nice to have a very clear statement in the intro and discussion about the eventual target of the drugs. The analysis was done with human data and considered aging and the analysis defines drugs that "reverse" the effect of SARS-CoV-2 infection, but in the context of an ACE inhibitor, so are the drugs targeting an infected patient or supposed to be used in a preventive setting. Are they only targeting lung tissue? And only in older people? I would like a bit more context of the ultimate purpose of the drugs given the results in the context of the data and analysis applied.

Our main goal was to identify FDA approved drugs that could be repurposed for COVID-19 patients. Given the increased morbidity and fatality rate of COVID-19 in ageing populations, we intersected the SARS-CoV-2 and ageing pathways to identify drugs that target proteins at their intersection. The identified drugs could be repurposed in the current crisis post-infection. The major drug targets identified through our analysis are serine/threonine and tyrosine kinases, which are critical intermediates in signaling pathways that get activated during epithelial to mesenchymal transitions in ageing lung tissues. In a recent perspective (Nature Reviews MCB 21 (2020), reference 21 in the manuscript), we conjectured that the ageing dependent mesenchymal cell states of the lung epithelium may facilitate viral replication. Based on the findings in the current manuscript, our hypothesis is that inhibiting these signaling pathways may reduce the aging dependent epithelial to mesenchymal transitions and may thereby reduce viral replication and pathogenesis. In particular, our analysis identified RIPK1, a serine/threonine protein kinase, and we suggest that inhibiting this kinase post-infection may reverse the effect of the virus. We now explicitly state in both the introduction and the discussion that the identified drugs and drug targets are of interest post-infection to reverse the effect of the virus.

Figure 6: Quantitative analysis of similarity between drug lists obtained using the latent space embedding as compared to the original and PCA embedding (using 2 PCs). Given two rankings of n drugs, we consider the top k drugs and plot the number of drugs in common among these top k drugs for $k = 1, 2, \dots, n$; i.e., the x -coordinate of a point indicates the proportion, k/n , of each drug list we consider and the y -coordinate is the size of the intersection of the two subsets normalized by k . AUC denotes the area under the curve; (a) shows the result when considering two maximally different drug lists, i.e., when one is the reverse of the other, resulting in an AUC of 0.307; (b) demonstrates that the drug list produced in the latent space of the over-parameterized autoencoder is similar to that produced in the original space and to that produced using 2 PCs. The advantages of using the over-parameterized autoencoder are that the resulting latent space contains enough signal to reconstruct gene expression vectors well and provides better alignment between drug signatures across cell types than in the original space.

Figure 7: Reformatted Fig. 2.

- Please add a limitations or caveats paragraph in the discussion to highlight a few of the limitations of this study. While the authors did show that the autoencoder can be applied to other cell lines, the study is fully based on the A549 cells, which is an inherent biological limitation. The other items mentioned above could be added to this paragraph as well.

We thank the reviewer for the suggestion. We added the following text in the discussion section: “While our method is broadly applicable, we note several limitations. First, our drug repurposing pipeline relies on the availability of RNA-seq data from normal and infected/diseased cells in the cell type of interest and therefore the availability of such data is necessary for the application of our platform. Second, since our autoencoder is trained on CMap data, which only contains the expression of 1000 genes (L1000 genes), it is possible that the signal of the infection may not be captured by these 1000 genes. However, this can be checked by assessing whether there is a statistically significant overlap between the L1000 genes and the differentially expressed genes in the disease of interest, which we performed in our analysis for SARS-CoV-2. Finally, since the CMap data contains a limited set of drugs, it is possible that none of the drugs are anticorrelated with the disease signature, thus preventing the user from identifying drug candidates.”

Minor Points

- Figure 2f, please label the columns; Figure 2g, add individual ranks as columns to see what the ranks are per experiment

We added ranks for Fig. 2g and column labels for Fig. 2f as suggested (see the revised figure in Fig. 7 above).

- Figure 3a, not needed, but it might help to show the genes that are L1000 and DE in a different color to give a sense to the number of genes that fall into both categories.

In the current Fig. 3a, the genes that are L1000 and DE correspond to stars that overlap with red circles.

Figure 8: Reformatted Fig. 4.

- Figure 4d is not readable with very tiny text, same with the gene names in the network figures.

We have now made the text more readable by providing it in vector format so that the reader can zoom in to read the gene names (see the revised figure in Fig. 8 above).

Reviewer #2 response:

The paper "Causal Network Models of SARS-CoV-2 Expression and Aging to Identify Candidates for Drug Repurposing" presents a computational systems medicine approach to drug target prediction for COVID-19. Essentially, the computational pipeline integrates transcriptomics, proteomics and structure data as well as drug screening data with biomolecular interaction networks. Interestingly, the authors performed the analyses such that aging-specific profiles are taken into account. The result is a candidate mechanism presumably highlighting serine/threonine as well as protein kinases as targets for inhibition.

Essentially, the authors identify genes differentially expressed in aging as well as SARS-CoV-2 infected cells. Through CMAP embedding, approved drugs are prioritized across different cell types and generalized for SARS-CoV-2. Finally, the authors run a Steiner tree analysis to extract age-specific and infection-specific PPI subnetworks, which are then intersected to extract a candidate mechanism, which is potentially enriched with druggable target genes driving age-dependent COVID-19 disease progression.

Along the the way many different computational tools are employed, all of which have different preconditions, advantages and (most importantly) require meta-parameters and cutoffs to be set. While a big part of the paper deals with justifications for each of the steps, one might have the impression that different tools and settings have been employed until a desired result was observed.

The paper in my opinion either needs (1) a preclinical / wet lab follow-up to demonstrate that the findings have clinically exploitable impact, (2) evaluation on completely independent data of different type(s) to backup the results/conclusions, or (3) an interactive software or web tool allowing other researchers to replicate the results and to visually explore the results for different algorithms and parameter choices - allowing others to bring in own background knowledge and to judge and modify the intermediate results properly.

We thank the reviewer for these helpful comments. As per the editor's suggestion, in the following we carefully address (2) and (3).

(2): In order to backup our results, we performed the analysis on two additional data sets for SARS-CoV-2 infection; namely, we considered A549 cells without ACE-2 supplement infected with SARS-CoV-2 and A549-ACE2 cells infected with lower MOI (0.2 instead of 2). Both datasets were obtained from [1], as was the data we performed the original analysis on. For both datasets, we obtained a list of drugs ranked by their correlation with the reverse disease signature based on our overparameterized autoencoder embedding. Fig. 3 above quantifies how similar these drug lists are to the original drug lists using a curve akin to an ROC curve; see our response to reviewer #1 for an in depth discussion. These figures show that the drug lists produced by considering other SARS-CoV-2 samples from [1] are very similar, indicating that the drug lists obtained by our analysis are robust. The similarity between the drug lists is the highest (AUC= 0.946) when comparing SARS-CoV-2 infection in A549 cells with and without ACE-2 supplement, both infected with MOI of 2. The similarity between the drug lists drops (AUC= 0.875 and 0.870) when comparing an MOI of 2 to an MOI of 0.2, which is consistent with the observation by [1] that low-MOI conditions did not stimulate an important interferon-I and -III response and thus may result in different suggested drugs.

To put these AUC values into perspective, we also repeated our analysis on two other viruses: respiratory syncytial virus (RSV) and influenza A virus (IAV). Gene expression data for this analysis was again obtained from [1]. Fig. 2 above shows that the drug lists obtained for different SARS-CoV-2 batches (on A549 cells with and without ACE-2 supplement and with MOI of 2 and 0.2) are more similar to each other than the drug lists obtained for IAV or RSV; see our response to reviewer #1 on this topic.

To reflect this analysis we added the following sentences in the revised main text: "To... assess the robustness of the identified drug list using the autoencoder embedding, we repeated the analysis on two other SARS-CoV-2 datasets from [1], namely infected A549 cells without ACE-2 supplement as well as samples collected at a lower MOI (0.2 instead of 2). This resulted in very similar drug lists (Supplementary Fig. S10); in fact the drug lists from A549 cells with and without ACE-2 supplement in the autoencoder embedding were more similar than the drug lists obtained from the PCA and the original space embedding."

(3): We now provide code for training the autoencoder and obtaining ranked drug lists as well as an interactive Jupyter notebook (Python 3) for the Steiner tree and causal analysis, which allows researchers to replicate all our results, reproduce the visualizations, as well as analyze how different hyperparameters affect the output. A ReadMe section is included in the notebook to provide information on the technical requirements and main steps of the analysis. The notebook includes a sensitivity analysis for the hyperparameters of the prize-collecting Steiner tree algorithm along with visualizations that allows the user to select a robust hyperparameter configuration. It also computes the list of drug targets and the corresponding drugs in the selected interactome, together with relevant metadata. A fine-grained analysis of the network can be performed by plotting the neighborhood of nodes of interest.

To reflect this, we added the following sentence in the code availability section in the revised manuscript: "Our code is available at http://github.com/uhlerlab/covid19_repurposing".

The presented work - although showing significance of the individual steps - at the very least also requires a proper significance analysis of the overall strategy. If the exact same data is used and the exact same thresholds, parameters, and algorithms, how would the results change when running on (1) randomized PPI networks with similar graph properties, (2) permuted expression data, (3) randomly selected CMAP signatures, (4) randomly picked genes instead of the Steiner-tree-selected ones, etc. The strategy would still provide results, but (hopefully) very different ones. How different? Would the "performance" drop? And how would "performance" be measured?

We thank the reviewer for this helpful suggestion. As proposed by the reviewer, we conducted several randomization assessments to understand the importance of each step in the pipeline, analyzing the impact of changes in the RNA-seq expression data, the underlying protein-protein interaction network, the CMap drug signatures, as well as the list of terminal genes on the final selection of drug targets and corresponding drugs. This was quantified by the frequency of appearance of each drug in the final drug list after 1000 randomization runs, for both drugs that were and that were not selected in the original non-randomized analysis. Results from this analysis suggest that the choice of terminal genes is the most

critical step of the Steiner tree procedure. Our results are summarized in Table 1 in this rebuttal (which is Table S2 in the revised Supplementary Materials).

To reflect this analysis, the following text was added in a new section on randomization analysis in the revised main text under Methods: "We conducted several randomization assessments to understand the importance of each step in the pipeline, analyzing the impact of changes in the RNA-seq expression data, the underlying protein-protein interaction network, the CMap drug signatures, as well as the list of terminal genes on the final selection of drug targets and corresponding drugs. This was quantified by the frequency of appearance of each drug in the final drug list after 1000 randomization runs, for both drugs that were and that were not selected in the original non-randomized analysis. Results from this analysis suggest that the choice of terminal genes is the most critical step of the Steiner tree procedure; see Supplementary Note and Supplementary Table S2."

In addition, the following text was added in the Supplementary Note describing the randomizations performed for this analysis:

(1) Randomization of PPI network: Randomization of the iREF protein-protein interaction network was performed via randomly permuting the vertex labels. Such randomization affects a gene's neighborhood while preserving basic network properties such as number of edges and degree distribution. The prize-collecting Steiner tree analysis pipeline was then applied to this new network. Drugs targeting terminal nodes were systematically selected in all randomization runs, as expected given that the prize-collecting Steiner tree algorithm parameters were set so that all terminal nodes are included in the solution. Other drugs identified by the non-randomized analysis that did not target any terminal node appeared with frequencies varying from 56% (primaquine, which has 5 targets in the network) to 97% (imatinib, which has 69 targets in the network). Only two drugs (mifepristone and palbociclib) that were not selected by the non-randomized analysis appeared more frequently (80% of runs) than the least frequently selected drug from the non-randomized analysis (primaquine, 56% of runs).

(2) Permuting expression data: Randomizing gene labels in the RNA-seq expression data set from [1] while preserving gene labels of the GTEx aging data set is an implicit approach to randomizing the list of terminal genes used as input for the prize-collecting Steiner tree algorithm. After applying the Steiner tree analysis pipeline, the drugs selected in the non-randomized analysis appeared between 18% (milrinone) and 100% (sunitinib) of the runs. Generally, the more proteins a drug targeted in the iREF network, the more frequently it appeared in the solution (sunitinib, with 260 targets, is the drug with highest number of targets in the PPI network). 16 drugs that were not selected in the non-randomized analysis (this represents 1% of the set of non-selected drugs) appeared more frequently than the least frequently selected drug from the non-randomized analysis (milrinone).

(3) Randomization of CMap signatures: We also ran the Steiner tree analysis after randomly permuting the SARS-CoV-2-anticorrelation scores of the 605 CMap drugs and selecting the drugs with anticorrelation above 0.86 (resulting in 142 drugs as in the original non-randomized analysis). After applying the Steiner tree analysis pipeline, drugs that were selected in the non-randomized analysis appeared in the final list with a frequency between 22% and 26%, as expected (since $142/605 \approx 23.5\%$). More interestingly, 17 drugs which were not selected in the non-randomized analysis (representing 1% of the overall set of non-selected drugs) appeared at a similar 22-29% frequency in the solution. These are drugs that target one of the network nodes yet have a true SARS-CoV-2-anticorrelation score lower than 0.86.

(4) Randomization of terminal nodes: Finally, we directly randomized the list of terminal nodes, by randomly selecting 162 genes from the RNA-seq expression dataset and prizing them with their corresponding absolute \log_2 fold change after SARS-CoV-2 infection in A549-ACE2 cells. The drugs selected in the non-randomized analysis appeared between 3% (milrinone) and 100% (sunitinib) of the runs. In this analysis, 41 drugs that were not selected in the non-randomized analysis (this represents 2.5% of the set of non-selected drugs) appeared more frequently than the least frequently selected drug from the non-randomized analysis (milrinone).

These results show that while the output of our Steiner tree analysis pipeline is quite robust to changes in the underlying PPI network, the selection of the terminal nodes has a critical effect on the final drug list."

drug	Selected	# targets in PPI	Frequency of appearance in randomizations			
			Gene labels	CMAP signatures	Terminal genes	PPI network
sunitinib	1	260	1.0	0.25	0.997	1.0
bosutinib	1	203	0.998	0.24	0.993	1.0
axitinib	1	99	0.997	0.25	0.98	1.0
dasatinib	1	128	0.98	0.246	0.98	1.0
sorafenib	1	116	0.998	0.266	0.975	1.0
pazopanib	1	103	0.991	0.235	0.965	1.0
ruxolitinib	1	132	0.988	0.243	0.94	1.0
erlotinib	1	96	0.967	0.234	0.933	1.0
afatinib	1	38	0.94	0.226	0.863	1.0
varденаfil	1	13	0.348	0.247	0.071	1.0
milrinone	1	9	0.178	0.253	0.034	1.0
imatinib	1	69	0.947	0.238	0.921	0.971
vorinostat	1	32	0.79	0.261	0.8	0.898
belinostat	1	11	0.743	0.225	0.755	0.867
docetaxel	1	13	0.422	0.251	0.576	0.796
tofacitinib	1	43	0.481	0.243	0.58	0.709
formoterol	1	5	0.326	0.253	0.499	0.59
primaquine	1	5	0.344	0.24	0.463	0.555
palbociclib	0	13	0.924		0.741	0.863
mifepristone	0	10	0.634		0.544	0.747
vemurafenib	0	4	0.246		0.393	
danazol	0	16	0.501		0.377	
tacrolimus	0	13	0.418		0.29	
haloperidol	0	42			0.286	
bicalutamide	0	2	0.278		0.277	
clozapine	0	39			0.195	
risperidone	0	36			0.188	
sulconazole	0	25			0.186	
econazole	0	41	0.439		0.164	
amitriptyline	0	33			0.138	
clemastine	0	25			0.103	
dipyridamole	0	19	0.353		0.103	
phentolamine	0	17			0.095	
iloperidone	0	24			0.092	
methysergide	0	22			0.092	
cyproheptadine	0	29			0.09	
carteolol	0	2			0.083	
lenalidomide	0	2			0.083	
cabergoline	0	17			0.079	
loxapine	0	29			0.079	
digitoxin	0	9			0.076	
terconazole	0	17	0.198		0.069	
ketotifen	0	17			0.065	
desipramine	0	22			0.054	
rosuvastatin	0	2			0.054	
perphenazine	0	16			0.053	
naftifine	0	2			0.05	
desoximetasone	0	1			0.048	
flunisolide	0	1			0.048	
halcinonide	0	1			0.048	
irinotecan	0	7			0.048	

phenelzine	0	10			0.048	
prednisone	0	2			0.048	
bupirone	0	13			0.046	
guanfacine	0	8			0.043	
terazosin	0	7			0.039	
sertraline	0	19			0.038	
flumazenil	0	36			0.037	
daunorubicin	0	1			0.036	
bortezomib	0	15		0.241		
caffeine	0	3	0.324			
cisplatin	0	10		0.234		
clofarabine	0	2	0.216			
dobutamine	0	23		0.226		
famotidine	0	3		0.24		
gefitinib	0	72		0.232		
glimepiride	0	4	0.18			
iloprost	0	8	0.206			
lapatinib	0	13		0.256		
midodrine	0	1		0.254		
mitoxantrone	0	18		0.23		
montelukast	0	21		0.251		
nilotinib	0	70		0.292		
olaparib	0	4		0.261		
panobinostat	0	11		0.247		
sildenafil	0	20		0.233		
sitagliptin	0	2	0.183			
tamoxifen	0	51		0.278		
tolbutamide	0	2	0.18			
topotecan	0	5		0.277		
treprostinil	0	6	0.206			
warfarin	0	1		0.254		
zafirlukast	0	13		0.238		

Table 1: Frequency of a drug’s presence in the list of final drugs after performing Steiner tree analysis with randomization of gene labels, CMap signatures, terminal genes, and the PPI network (1000 randomization runs).

In addition to these randomization experiments and the application to two additional SARS-COV-2 datasets described above, we also performed the following additional robustness check. The CMap dataset is known to contain various batch effects, which we had identified and mitigated by k-means clustering on the control samples and removal of the smaller cluster (see Supplementary Figure S6). Further analysis of the removed cluster showed that the dropped cluster consisted of samples whose minimum gene expression value was 1 (after $\log_2(x + 1)$ scaling), while all other gene expression values fell in the range of [5, 13]. To ensure robustness of our results to different ways of mitigating batch effects in the CMAP dataset, we repeated the analysis by dropping all genes for which there was at least one sample containing a 1 in the expression value (reducing the total number of genes from 911 to 867 for the A549 cell line). As with the original batch correction approach, the resulting drugs consist mainly of protein kinase inhibitors (7 out of 9) and the drug targets are highly overlapping with the drug targets obtained from the original analysis; see Fig. 9 below (which is Supplementary Fig. S24 in the revised Supplementary Materials).

To reflect this additional analysis, we added the following sentences in the revised Methods section of the manuscript: "Subsequent analysis of the removed cluster revealed that it consisted of samples with a minimum gene expression value of 1 (after $\log_2(x + 1)$ scaling), while all other gene expression values fell in the range of [5, 13], thereby providing further reason for removal of this cluster... to ensure robustness of our results to different ways of mitigating batch effects in the CMAP dataset, we repeated

the analysis by dropping all genes for which there was at least one sample containing a 1 in the expression value (reducing the total number of genes from 911 to 867 for the A549 cell line). As with the original batch correction approach, the resulting drugs consist mainly of protein kinase inhibitors (7 out of 9) and the drug targets are highly overlapping with the drug targets obtained from the original analysis (Supplementary Fig. S24).”

Figure 9: Comparison of drug targets resulting from analyzing the CMap dataset with and without removing confounding 1s.

Reviewer #3 response:

The authors developed a platform that systematically integrates available transcriptomic, proteomic and structural data to identify robust druggable protein targets. Their results highlight the importance of RIPK1 in the interplay between SARS-CoV-2 infection and aging as a potential target for drug repurposing programs. The methods and results reported are interesting especially that they integrate aging signatures into drug discovery platforms, which is very novel. I have several major concerns about this work.

1) Though "Rigorous in vitro experiments as well as clinical trials are needed to validate the identified candidate drugs" is declared. I think the authors need to do this validation themselves(or by cooperation) to make a full story.

We thank the reviewer for the thoughtful feedback. As per the editor's suggestion, we here concentrate on points 2)-4).

2) It seems that a pipeline of methods and datasets are used and integrated. Are there any novelty in the algorithms? I cannot figure out if any new algorithms are developed or customized to problems studied. Though figure 1 shows the overview of computational drug repurposing platform for COVID-19. I suggest a schematic diagram for the computational methods and how they were connected is required.

We thank the reviewer for the suggestion to add a schematic diagram of the inputs, outputs, and connections between the different computational methods in our drug repurposing pipeline, which also enables us to emphasize the novelty of our work. Such a schematic is shown below in Fig. 10 and also included in the revised Supplementary Materials (Supplementary Fig. S1).

Prior computational approaches have mainly focused on signature matching and analyzing disease networks in order to repurpose existing drugs [2]. Our work is novel in that we not only combine both

of these approaches but, most importantly, also innovate each of these methodologies. In particular, in terms of signature matching we use the novel framework of overparameterized autoencoders (autoencoders where the "bottleneck" is larger than the input dimension). We show that such autoencoders learn a latent embedding that better aligns drug signatures across cell types and at the same time provides better reconstruction than usual autoencoders. This allows us to predict the effect of a drug on a cell type without measuring it by using other cell types to infer it. Additionally, we improve upon the typical use of disease networks to identify drug mechanisms by learning a *causal* graph instead of merely an undirected network, which enables us to identify the downstream effect of a drug.

3) The authors claimed that the drug discovery platform is broadly applicable. Can they give a minimal requirement for the input data to the users? I mean more discussion on the methods and algorithms is needed.

We thank the reviewer for this question, which we mostly addressed in Fig. 10 (which corresponds to Supplementary Fig. S1) by representing the input data required by a user as green boxes. In particular, our drug repurposing pipeline requires RNA-seq samples of normal and diseased cells. If one is interested in including the effect of additional conditions (in our case aging), RNA-seq samples for those conditions are required as well. While for the first two parts of the pipeline (mining relevant drugs and identifying the disease interactome), only few RNA-seq samples are required (these could be from bulk data as in our application), the final part of the pipeline (investigating drug mechanism) requires the number of samples to be of the order of the number of nodes in the causal network. In particular, one may use single-cell RNA-seq data for this step, as we did in our application. All other inputs to the pipeline, namely CMap, DrugCentral and the PPI network (e.g. iREF or STRING), are available from public databases.

To better explain the methods and inputs required, we added the following text in the revised Supplementary Materials (Supplementary Note, Overview of Methods section): "Our drug discovery pipeline consists of three parts: mining relevant drugs, identifying the disease interactome, and investigating the drug mechanism. Fig. S1 describes the inputs, outputs and algorithms used in each of the three parts. Briefly, the first part (mining relevant drugs) takes in normal and infected/diseased RNA-seq samples along with the public CMap database, which contains gene expression data on cell lines treated with a variety of FDA approved compounds, to train an autoencoder and subsequently construct synthetic interventions in the learned latent space. It outputs a list of drugs ranked by the correlation of each drug with the reverse disease signature. The second part of the pipeline (identifying disease interactome) also takes in the normal and infected/diseased RNA-seq samples as well as a PPI network (e.g. from the public iREF or STRING databases). It then identifies the genes that are differentially expressed in the disease and learns the disease interactome connecting these genes in the PPI network using the prize-collecting Steiner forest algorithm. In addition, the inferred ranked list of drugs output from part 1 in the pipeline is mapped to its targets using the public DrugCentral database. The drug targets are intersected with the disease interactome to further filter the list of drugs to only include those drugs that target nodes in the interactome. The third part of the pipeline (investigating drug mechanism) uses multi-sample RNA-seq data (e.g. high number of replicates or single-cell RNA-seq data) to learn the causal directions in the disease interactome using GSP, a causal structure discovery algorithm, and identifies which drugs and drug targets have the largest downstream causal effect on the disease interactome."

We also clarified the minimal requirement for the input data in the discussion section of our revised manuscript: "our drug repurposing pipeline relies on the availability of RNA-seq data from normal and infected/diseased cells in the cell type of interest and therefore the availability of such data is necessary for the application of our platform."

4) It is really interesting that the importance of RIPK1 in the interplay between SARS-CoV-2 infection and aging as a potential target for drug repurposing programs was found. Can the authors give more discussion on and insight into what is the difference between young and old individuals after SARS-CoV-2 infection besides result representation.

The major targets of the identified drugs are serine/threonine as well as tryosine kinases, which are critical intermediates in signaling pathways that get activated during epithelial to mesenchymal transitions in ageing lung tissues. In a recent perspective (Nature Reviews MCB 21 (2020), reference 21 in

Figure 10: Detailed schematic of our computational drug repurposing platform. Green boxes denote inputs that may need to be collected for the specific virus/disease and cell type of interest. Blue boxes denote inputs corresponding to databases that are publicly available. Orange boxes denote our computational methods and yellow boxes denote method outputs. Computational pipeline for (a) mining relevant drugs, (b) identifying disease interactome and (c) investigating drug mechanism.

the manuscript), we conjectured that the ageing dependent mesenchymal cell states of the lung epithelium may facilitate viral replication. Based on the findings in the current manuscript, our hypothesis is that inhibiting these signaling pathways may reduce the aging dependent epithelial to mesenchymal transitions and may thereby reduce viral replication and pathogenesis. In particular, our analysis identified RIPK1, a serine/threonine protein kinase, and we suggest that inhibiting this kinase post-infection may reverse the effect of the virus.

References

1. Blanco-Melo, D. *et al.* Imbalanced host response to SARS-CoV-2 drives development of COVID-19. *Cell* (2020).
2. Pushpakom, S. *et al.* Drug repurposing: progress, challenges and recommendations. *Nature Reviews Drug Discovery* **18**, 41–58 (2019).

Reviewers' Comments:

Reviewer #1:

Remarks to the Author:

The authors took the reviewers comments seriously and performed additional analysis and included these analyses mostly in the supplement. The authors addressed my concerns and did a nice job addressing all reviewer comments. I have no further concerns.

Reviewer #2:

None

Reviewer #1 response:

The authors took the reviewers comments seriously and performed additional analysis and included these analyses mostly in the supplement. The authors addressed my concerns and did a nice job addressing all reviewer comments. I have no further concerns.

We thank the reviewer for their positive comments.